



# A novel algorithmic framework for identifying changing streamflow regimes: Application to Canadian natural streams (1966-2010)

Masoud Zaerpour[1], Shadi Hatami[1], Javad Sadri[2] and Ali Nazemi[1]

[1]Department of Building, Civil and Environmental Engineering, Concordia University, Montreal, Quebec, Canada

[2]Oppimi Group, Montréal, Quebec, Canada

*Correspondence to*: Masoud Zaerpour (masoud.zaerpour@concordia.ca)

**Abstract.** Climate change significantly affects natural streamflow regime. To assess alterations in streamflow regime, typically few streamflow characteristics are considered and their significant variations in time and space are taken as a notion of change. Although, this approach is informative, intuitively appealing and widely-implemented, (1) it cannot see simultaneous changes

in multiple streamflow characteristics; (2) it does not utilize all the available information contained in a streamflow hydrograph; and (3) it cannot describe how and to what extent one streamflow regime evolves into other regime types. To address these gaps, we conceptualize streamflow regimes as intersecting spectrums that are formed by multiple streamflow characteristics. Accordingly, we recognize that changes in streamflow regime should be diagnosed through gradual, yet continuous changes in an ensemble of streamflow characteristics. To incorporate these key considerations, we propose a fuzzy clustering-based

approach to classify the natural streamflow into a finite set of intersecting regime types. Accordingly, by analyzing how the degrees of membership to regime types change, we quantify monotonic shifts between regime types in time and space. Our proposed algorithm eliminates the subjectivity in quantifying shift between flow regimes, and can extract valuable knowledge stored in the shape and variability of annual streamflow hydrographs. We apply this approach to the natural streamflow data, obtained from 106 Canadian gauges, during the period of 1966 to 2010. We show that natural streamflow in Canada can be

categorized into six regime types, with clear physical and geographical distinctions. Analyses of trends in membership values during the study period show that alterations in natural streamflow regime are vibrant and can be different within and between major Canadian drainage basins. We show that gradual changes in natural streamflow regimes in Canada can be attributed to simultaneous changes in a large number of streamflow characteristics, some of which have been previously unknown or not well-attended. Our study introduces a generic algorithmic framework for identifying changing streamflow regime at regional

and global scales, and provides a fresh look at streamflow alterations in Canada, which can be seen as another line of evidence for the complex and multifaceted impacts of climate change on streamflow regime, particularly in cold regions.

## 1 Introduction

Natural streamflow characteristics have been critical consideration for ecosystem and human developments around rivers, globally (Hart and Finelli, 1999; Poff et al., 2010; Nazemi and Wheater, 2014; Hassanzadeh et al., 2017). For instance, since



early settlements, human learned that timing and duration of low flows control river biodiversity, riparian vegetation and water quality (Rolls et al., 2012; Ireson et al., 2015; Knouft and Ficklin, 2017). During the current *"Anthropocene"*, streamflow characteristics are key factors for infrastructure design as well as land use and land management. While some streamflow characteristics reveal potentials for natural resource development, particularly for agriculture and hydropower production (Hamududu and Killingtveit, 2012; Amir Jabbari and Nazemi, 2019; Nazemi et al., 2020), some others determines consequences of important natural disasters such as flood and drought (Poff and Zimmerman, 2010; Arheimer and Lindström, 2015; Burn and Whitfield, 2016; Rolls et al., 2018; Zandmoghaddam et al., 2019). A set of natural streamflow characteristics determining timing, magnitude, seasonality and inter-annual variability in streamflow time series can collectively define the streamflow regime (Poff et al., 1997). Traditionally, streamflow regimes are considered stationary in time (Milly et al. 2005, 2008, 2015). However, the looming effects of climate change along with massive human interventions through land and water management have raised fundamental questions against the feasibility of stationarity assumption for streamflow conditions (Döll and Zhang, 2010; Arnell and Gosling, 2013; Nazemi and Wheater, 2015a, 2015b; Döll et al., 2018). The contemporary literature is full of evidences, revealing major alterations in natural streamflow regime in various regions, induced by heightened climate variability and change (Barnett et al., 2005; Stahl et al., 2010; Rood et al., 2016; Blöschl et al., 2017; Hodgkins et al., 2017; Nazemi et al., 2017; Dierauer et al., 2018). Moreover, projections of future streamflow conditions under climate change conditions show significant alterations in streamflow regime globally in years to come (Zhang et al., 2016; Asadieh, and Krakauer, 2017; Eisner et al., 2017; Gizaw et al., 2017; Grantham et al., 2018).

Climate change impacts on natural streamflow regime are more severe in higher latitudes such as in Canada (e.g., Nijssen et al., 2001; Déry and Wood, 2005; Hinzman et al., 2005; Leclerc and Ouarda, 2007; IPCC, 2013; DeBeer et al., 2016; Brahney et al., 2017; MacDonald et al., 2018; Islam et al., 2019; Champagne et al., 2020 Dierauer et al., 2020), where the rate of warming is twice of the global average (Bush and Lemmen, 2019). Both observed and projected changes in Canadian streamflow characteristics are subject to significant spatial variabilities (e.g., Burn et al., 2010; Buttle et al., 2016; O'Neil et al., 2017). In northern Canada, for instance, an increase in spring runoff is expected; whereas in southern Pacific, decreasing summer flows are projected (Kang et al., 2016; Curry et al., 2019; Islam et al., 2019). These differences are not only between different regions and/or drainage basins, but can be observed within the same basin and/or between two tributaries with relatively close proximity. For instance, there are significant differences between forms of change in streamflow regime between northern and southern parts of the Pacific (Déry et al., 2009; Monk et al., 2011; Kang et al., 2016; Brahney et al., 2017). Similarly, in northern Canada, glacier-fed rivers show increases in summer runoff (Fleming and Clarke, 2003; Stahl and Moore, 2006); whereas other regime types show tendency toward decreasing summer runoff (Fleming and Clarke 2003; Hinzman et al., 2005; Janowicz, 2008, 2011; Foy et al., 2011).

Despite the body of knowledge already gathered around assessing the effects of climate change on streamflow regime, there are still rooms for methodological improvements. Most importantly, among many potential flow characteristics that can constitute and describe streamflow regime, often only few have been taken into account, not only in the Canadian context, but also globally (Whitfield and Cannon, 2000; Regonda et al., 2005; Stewart et al., 2005; Maurer et al., 2007; Blöschl et al.,



2011, 2017; Hall et al., 2014; Vormoor et al., 2015). This is indeed a technical limitation, because climate change impacts are
often manifested in the entire streamflow hydrograph, and not only around certain streamflow characteristics (Olden and Poff,
2003). This is due to the fact that at the watershed scale, multiple streamflow generation mechanisms are involved that behave
differently in response to climate variability and change (Whitfield and Pomeroy, 2016). This is particularly the case in cold
regions, where alterations in the streamflow regime are formed due to compound impacts of changes in temperature, forms
and magnitude of precipitation, as well as melting snowpack and glacial storages (DeBeer et al., 2016; Curry and Zwiers,
2018; Hatami et al., 2018; Glas et al., 2019; Rottler et al., 2020). At this stage of development, it is not entirely clear how
changes in streamflow regime can be quantified using a large set of streamflow characteristics that together represent expected
annual streamflow hydrograph, as well as its inter-annual variability (Burn et al., 2016; Burn and Whitfield, 2018).

Here, we propose a new methodology to address this challenge. We recognize that by considering more streamflow
characteristics, the distinctions between regime types and their forms of alterations become less apparent and more relative.
Accordingly, in line with some recent suggestions in the literature (see e.g., Ternynck et al., 2016; Burn and Whitfield, 2017;
Knoben et al., 2018; Brunner et al., 2018, 2019; Aksamit and Whitfield, 2019; Jehn et al., 2020), we fundamentally
conceptualize streamflow regime as a continuous spectrum rather than some rigidly defined and distinct states. This
conceptualization requires a methodology that can formally deal with relativity in the definition of streamflow regime and its
changes in time and space. For this purpose, we use elements of fuzzy set theory (see Zadeh, 1965) to provide the
methodological basis to classify streamflow regimes as fuzzy clusters rather than clear-cut regime types. We then measure the
gradual departure from one fuzzy cluster to others using observed monotonic trends in membership degrees, and use this
information as an indicator of regime shift. This provides a systematic approach for quantification of shifts between flow
regimes. Accordingly, we highlight how gradual shifts in regime types are attributed to changes in streamflow characteristics.
By implementing this algorithm to more than 100 natural streams in Canada during a unified period, we provide a
homogeneous, pan-Canadian view on recent alterations in natural streamflow regime across the country. The remainder of this
paper is structured as the following: Section 2 describes our three-phase methodology related to (i) clustering of regime types,
(ii) detection of regime change, and (iii) attributions of changes to streamflow characteristics. Section 3 briefly introduces the
natural Canadian streamflow data used in this study. The results and discussions are presented in Sects 4 and 5, respectively.
Finally, Sect. 6 concludes our work and provides some further remarks.

## 2 Methodology

### 2.1 Rationale and proposed algorithm

Here, we aim at building a multistep algorithm to (1) classify natural streamflow regimes into a finite set of interpolating
regime types, (2) diagnose the gradual evolution in regime types and their shift from one type to another using a systematic
measure, and (3) attribute the changes in streamflow regime to alterations in a specific set of streamflow characteristics. The
proposed algorithm is built upon two fundamental considerations. First, we acknowledge that streamflow regimes are





constituted by several streamflow characteristics, and therefore changes in streamflow regimes should be also manifested through changes in an ensemble of streamflow characteristics. Second, we recognize that there are soft rather than hard distinctions between streamflow regimes, and regime shifts occur rather gradual than abrupt. These two considerations lead to development of our proposed algorithm to classify the streamflow regime using a fuzzy clustering approach and monitoring

the gradual variations in the degree of belongingness to each cluster in time and space. In brief, we select a set of streamflow characteristics (or features) to collectively characterize the streamflow regime – see Sect. 2.2. We then use the Fuzzy C-Means algorithm (FCM), which is a well-known clustering approach, to classify streams into a set of overlapping regime types during a common baseline period – see Sect. 2.3. We accordingly quantify changes in degrees of association to each regime type during the entire data period using a moving trend analysis. By monitoring the co-occurrence of divergent trends in membership

values, the transitions of regime types to one another can be identified – see Sect. 2.4. Finally, we monitor the co-evolution of regime shifts with the alterations in streamflow characteristics through a formal dependency analysis – see Sect. 2.5. Figure 1 shows the proposed three-step procedure. Below we describe each step of this algorithm in more details.

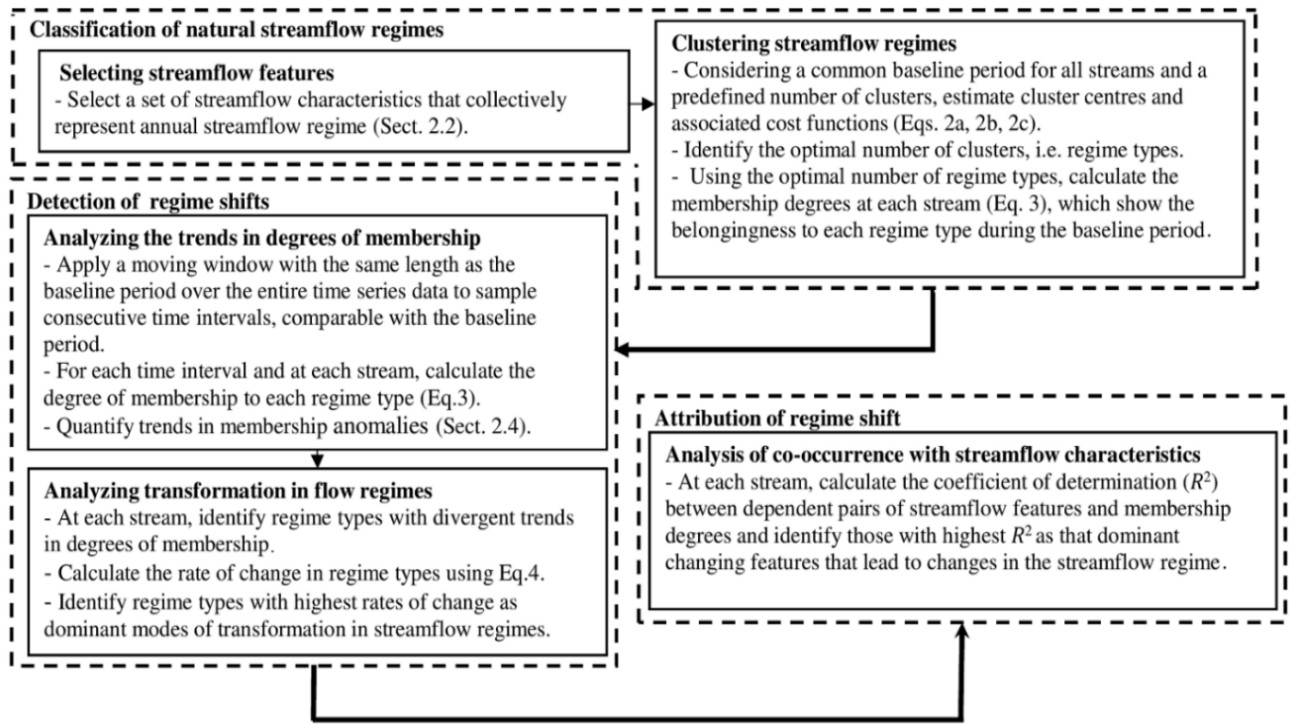

**Figure 1.** The workflow of the proposed algorithm for classifying streamflow regime, diagnosing shift in streamflow regime, and attributing the regime shift to changes in streamflow characteristics.



## 2.2 Feature selection

Indicators of Hydrologic Alterations (IHAs: Richter et al., 1996) are a set of streamflow characteristics that are commonly
applied as *features* to characterize changes in natural streamflow regime (e.g., Hu et al., 2008; Yang et al., 2012; Wang et al.,
2018). Different set of IHAs can be considered depending on the application in hand. As the application presented in this paper
concerns annual streamflow regime in Canada, see Sect. 3, we consider 15 IHAs, including annual mean flow, monthly mean
flows as well as timings of the annual low and high flows. At each stream, we use the mean (first moment) and variance
(second moment) of these 15 indicators during a multi-year timeframe to come up with the 30 features that together can capture
the shape of expected annual hydrograph and its inter-annual variability during the considered timeframe. Table 1 shows the
name and notation of streamflow features used in this study, where $x_{j=1:15}$ and $y_{j=1:15}$ correspond to mean and variance of the
15 IHAs, respectively.

**Table 1.** The thirty streamflow features used for clustering natural streamflow regime in Canada.

| Feature | Notation | Feature | Notation | Feature | Notation | Feature | Notation | Feature | Notation |
|---|---|---|---|---|---|---|---|---|---|
| October mean flow | mean: $x_1$ variance: $y_1$ | November mean flow | mean: $x_2$ variance: $y_2$ | December mean flow | mean: $x_3$ variance: $y_3$ | January mean flow | mean: $x_4$ variance: $y_4$ | February mean flow | mean: $x_5$ variance: $y_5$ |
| March mean flow | mean: $x_6$ variance: $y_6$ | April mean flow | mean: $x_7$ variance: $y_7$ | May mean flow | mean: $x_8$ variance: $y_8$ | June mean flow | mean: $x_9$ variance: $y_9$ | July mean flow | mean: $x_{10}$ variance: $y_{10}$ |
| August mean flow | mean: $x_{11}$ variance: $y_{11}$ | September mean flow | mean: $x_{12}$ variance: $y_{12}$ | Annual flow | mean: $x_{13}$ variance: $y_{13}$ | Timing of the annual low flow | mean: $x_{14}$ variance: $y_{14}$ | Timing of the annual high flow | mean: $x_{15}$ variance: $y_{15}$ |

## 2.3 Fuzzy C-means clustering

Clustering is the process of arranging data into a finite set of classes, in a way that members in the same class have similar
characteristics. The statistical methodologies used for clustering in hydrology are numerous (see Monk et al., 2011; Olden et
al., 2012; Ternynck et al., 2016; Tarasova et al., 2019; Wolfe et al., 2019; Brunner et al., 2020) and have been traditionally
limited to non-overlapping (i.e. hard) classes. Recent theoretical developments have relaxed this assumption and considered a
set of overlapping (i.e. soft) classes, in particular in the form of fuzzy sets. The association to each fuzzy cluster can quantified
using a degree of belongingness, also known as the membership value (see Bezdek, 1981; Sikorska et al., 2015, Piniewski,
2017; Knoben et al., 2018). The process of FCM for clustering streamflow regime can be summarized as the following: Assume
that flow data of $N$ streams during a common timeframe $w$ with length of $l$ years are available. For each stream, first and
second moments of $n$ IHAs (here $n = 15$), i.e. $\mathbf{X} = [x_{ij}]$, $\mathbf{Y} = [y_{ij}]$; $i \in \{1, ..., N\}, j \in \{1, ..., n\}$, can be extracted during the
considered timeframe $w$. Accordingly, the extracted features can be normalized to avoid scale mismatches in the feature matrix:




$$\bar{x}_{i,j} = \frac{x_{i,j} - \min\{x_{i=1:N,j}\}}{\max\{x_{i=1:N,j}\} - \min\{x_{i=1:N,j}\}} \quad \forall j \in \{1,\ldots,n\} \tag{1a}$$

$$\bar{y}_{i,j} = \frac{y_{i,j} - \min\{y_{i=1:N,j}\}}{\max\{y_{i=1:N,j}\} - \min\{y_{i=1:N,j}\}} \quad \forall j \in \{1,\ldots,n\} \tag{1b}$$

where $\bar{\mathbf{X}} = [\bar{x}_{ij}]$ and $\bar{\mathbf{Y}} = [\bar{y}_{ij}]$ are the matrices of Normalized Streamflow Features (NSFs). FCM partitions the $N$ streams into $c$ fuzzy clusters, such that the sum of distances for all streams $i \in \{1,\ldots,N\}$ between normalized feature vector and cluster

centroids is minimized. This is through an iterative procedure, which aim at finding the cluster centroid by minimizing the following objective function:

$$J(\mathbf{U},\mathbf{V}\,|\bar{\mathbf{X}},\bar{\mathbf{Y}}) = \sum_{k=1}^{c}\sum_{i=1}^{N}(u_{i,k})^2 d^2\left([\bar{x}_{i,j=1:n}\bar{y}_{i,j=1:n}], v_{k,m=1:2n}\right) \tag{2a}$$

This objective function is subject to the following two constraints:


$$\sum_{k=1}^{c} u_{i,k} = 1 \quad \forall i \in \{1,\ldots,N\} \tag{2b}$$

$$0 < \sum_{i=1}^{N} u_{i,k} < N \quad \forall k \in \{1,\ldots,c\} \tag{2c}$$

where $v_{k=1:c,m=1:2n} = \left[\overline{x^*}_{k,j=1:n}\overline{y^*}_{k,j=1:n}\right] = \left[\overline{x^*}_{k,1},\ldots,\overline{x^*}_{k,n},\overline{y^*}_{k,1},\ldots,\overline{y^*}_{k,n}\right] \in \mathbb{R}^{2n}$ is the matrix of cluster centroids (i.e. regime types); the matrix of $\mathbf{U} = [u_{i,k}]; i \in \{1,\ldots,N\}, k \in \{1,\ldots,c\}$ is the matrix of memberships; $\mathbf{V} = [v_{k,m}]; k \in \{1,\ldots,c\}, m \in \{1,\ldots,2n\}$ is the matrix of cluster centroids and $d^2\left([\bar{x}_{i,j=1:n}\bar{y}_{i,j=1:n}], v_{k,m=1:2n}\right)$ is the squared Euclidian

distance between matrix of normalized streamflow features and clusters' centroids. The fuzzy membership matrix can be calculated as:

$$u_{ik} = \frac{\left(\dfrac{1}{d^2\left([\bar{x}_{i,j=1:n}\bar{y}_{i,j=1:n}], v_{k,l=1:2n}\right)}\right)}{\sum_{k=1}^{c}\left(\dfrac{1}{d^2\left([\bar{x}_{i,j=1:n}\bar{y}_{i,j=1:n}], v_{k,l=1:2n}\right)}\right)}; \quad i \in \{1,\ldots,N\}, k \in \{1,\ldots,c\} \tag{3}$$





The number of clusters $c$ (i.e. regime types) can be chosen as a *priori* or empirically using validity indices (e.g., Pal and
Bezdek, 1995; Halkidi et al., 2001; Srinivas et al., 2008). Here we implement three validity indices of Xie-Beni index (Xie and
Beni, 1991; hereafter $V_{XB}$,), separation index (Bensaid et al., 1996; hereafter $V_S$,), and partition index (Bensaid et al., 1996;
hereafter $V_{SC}$) to come up with an optimal number of clusters.

### 2.4 Detection of change in streamflow regimes

Categorizing natural streams into $c$ regime types takes place in a baseline timeframe with the length of $l$ years, in which optimal
number of clusters, cluster centroids and the initial membership degrees to each regime type are identified. For each stream,
the timeframe can be moved year-by-year and the membership values can be recalculated for the new timeframe using Eq. (3).
This results into $c$ time series of membership values at each stream, showing how the association to each regime type evolves
at each stream – see Nazemi et al. (2017) and Jaramillo and Nazemi (2018) for more details on moving window methodology.
Figure 2 exemplifies this process in a hypothetical case. In order to quantify the gradual change in membership degrees, the
Mann-Kendall trend test with the Sen's Slope estimator (Mann, 1945; Sen, 1968; Kendall, 1975; Hamed, 2008; Gocic and
Trajkovic, 2013) can be applied to membership series.

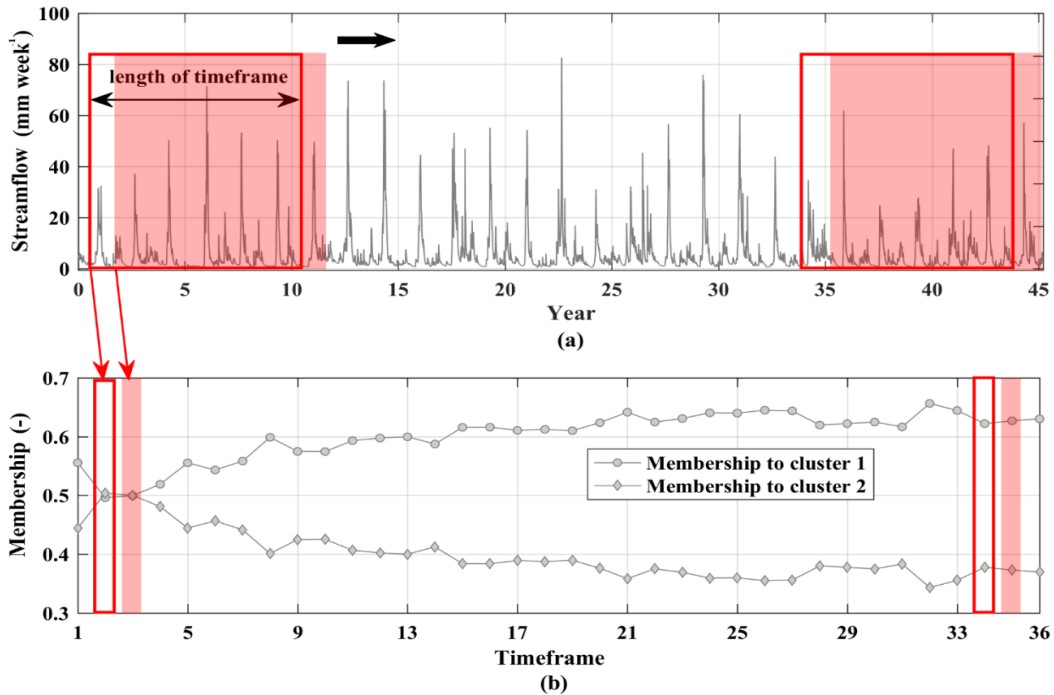

**Figure 2**. A schematic view to the procedure of identifying the evolution in membership values using a moving window; (a) a decadal
timeframe slides over the streamflow time series year-by-year; (b) membership degrees are recalculated at each decadal timeframe to
systematically determine the changes in association to each regime type determined in the beginning of the data period.





As the sum of memberships in each timeframe is one, gradual increase in memberships to one cluster should coincide with gradual decrease in the membership of one or more clusters. This transition can be identified by significant negative

dependencies between membership degrees to two clusters at each stream. Assuming the pair of clusters $p$ and $q$ in stream $i$, the rate of shift from $p$ to $q$ can be quantified as:

$$S_{i,(p,q)} = \left| \frac{\sum_{w=1}^{m} \left( u_{i,q}(w) - \mathbf{E}(u_{i,q}) \right) \left( u_{i,p}(w) - \mathbf{E}(u_{i,p}) \right)}{\sum_{w=1}^{m} \left( u_{i,q}(w) - \mathbf{E}(u_{i,q}) \right)^2} \right| \tag{4}$$

for $1 \leq w \leq m$ and $1 \leq p, q \leq c$ and $p \neq q$   $\forall\, i \in \{1, \ldots, N\}$

where $u_{i,p}(w)$ and $u_{i,q}(w)$ are membership degrees to clusters $p$ and $q$ in stream $i$ during the timeframe $w$; $w \in \{1, \ldots, r\}$; $r$

is the number of timeframe; $\mathbf{E}(u_{i,p})$ and $\mathbf{E}(u_{i,q})$ are the expected memberships; and $S_{i,(p,q)}$ is the slope of the best fitted line.

**2.5 Attribution of change in memberships to streamflow features**

Changes in membership degrees to each regime type can be attributed to changes in streamflow characteristics. Here, we recognize that the existence of significant dependence between membership values and streamflow features can provides a notion of attribution. Accordingly, we use the Kendall's tau (Genest and Favre, 2007; Nazemi and Elshorbagy, 2012) to detect

the co-occurrence between changes in memberships and changes in streamflow features. Figure 3 shows the procedure of attributing changes in membership values to changes in streamflow characteristics. Panels in the left column show the changes in membership degrees of two hypothetical clusters (purple lines), along with the corresponding changes in two normalized streamflow features (grey lines). Panels in the right column show the scatter plots of each time series of membership degrees versus the corresponding normalized streamflow features. The direction of dependence can be identified using the sign of

Kendall's tau coefficient. To measure the level of attribution between the change in streamflow features $x_{i,j}$ and membership values $u_{i,k}$, the coefficient of determination ($R^2$; see Legates and McCabe Jr., 1999) is used. $R^2$ varies within [0, 1] and determines how much the changes in the degrees of membership can be described by changes in streamflow characteristics. The higher $R^2$ is, the larger is the association between changes in the degrees of membership and the considered streamflow characteristics. The coefficient of determination in this case can be calculated as:


$$R^2(u_{i,k}, x_{i,j}) = \frac{\left\{ \sum_{w=1}^{r} \left( u_{i,k} - \mathbf{E}(u_{i,k}) \right) \left( x_{i,j} - \mathbf{E}(x_{i,j}) \right) \right\}^2}{\sum_{w=1}^{r} \left( u_{i,k} - \mathbf{E}(u_{i,k}) \right)^2 \sum_{w=1}^{r} \left( x_{i,j} - \mathbf{E}(x_{i,j}) \right)^2} \quad \forall\, i \in \{1, \ldots, N\} \tag{5}$$





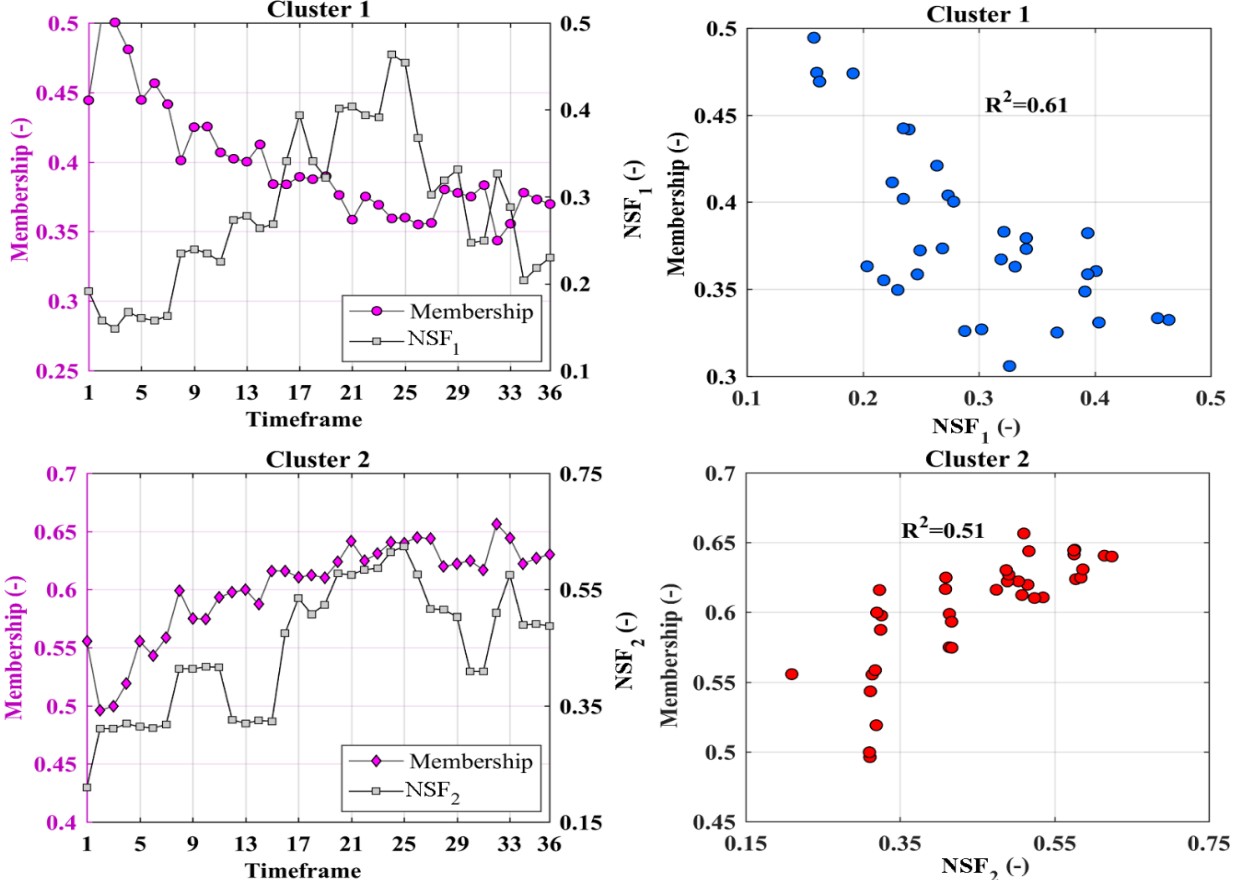

**Figure 3.** The procedure of attributing changes in membership degrees to changes in streamflow characteristics. The left column shows the co-evolution of membership degrees and Normalized Streamflow Features (NSFs). The right column measures the correspondence between changes in membership degrees and normalized streamflow features through percentage of described variance quantified using $R^2$. Red or blue dots show the positive or negative dependencies, respectively.

## 3 Case study and data

Canadian streamflow network is distributed across four major ocean-drained basins, namely Pacific, Atlantic, Arctic and Hudson Bay that together cover around 99.7% of Canada's surface area (Natural Resources Canada, 2007). The Pacific basin, spreading south to north from the US border to Yukon in the west coast, drains the total area of around 1 million km$^2$ into the Pacific Ocean. The basin is divided from the Arctic basin by the Canadian Rockies. The main sub-basins in the Pacific include Fraser, Yukon, Columbia and the Seaboard. In the east coast, the Atlantic basin drains the total area of 1.6 million km$^2$ to the Atlantic Ocean. This drainage basin includes important water bodies such as the Great Lakes and is mainly dominated by the streamflow regime in the St. Lawrence River and Seaboard, which are significantly larger sub-basins compared to the Saint





John-St. Croix. Towards north, the Arctic basin drains northern parts of Alberta, British Columbia and Saskatchewan, parts of Yukon as well as the Northwest Territories and the Nunavut. Having the drainage area of over 3.5 million km$^2$, the Arctic basin includes some of Canada's largest water bodies such as the Slave, Athabasca and Great Bear lakes as well as Peace, Mackenzie, and Lidar rivers. Mackenzie, Peace-Athabasca and Seaboard are the main sub-basins in the Arctic drainage basin. With the area of 3.8 million km$^2$, the Hudson Bay is the largest drainage basin in Canada, covering five provinces from Alberta

in the west to Quebec in the east. The basin includes four major sub-basins, namely Western & Northern Hudson Bay, Nelson, Northern Ontario, and Northern Quebec. Nelson, Saskatchewan and Churchill rivers are the major river systems in the Hudson Bay (Pearse et al., 1985; Statistics Canada, 2009). For the sake of convenience, in this paper we consider Northern Ontario and Northern Quebec sub-basins as one single assessment unit.

      We use the data from Reference Hydrometric Basin Network (RHBN; Water Survey of Canada, 2017,

http://www.wsc.ec.gc.ca/) to diagnose simultaneous changes in natural streamflow regimes across Canadian major basins. RHBN includes 782 stations that measures streamflow at unregulated tributaries over Canada, and therefore are particularly suitable to address climate change impacts on natural streamflow regime (Brimley et al., 1999; Harvey et al., 1999; Whitfield et al., 2012; Zadeh et al., 2020). Considering the available data for the period of 1903 to 2015, we searched for the largest subset of stations with longest continuous daily record during a common period with negligible missing data (i.e., less than a

month in a typical year). This results into selecting 106 streamflow gauges during the water years of 1966 to 2010 (1 October 1965 to 30 September 2010). Table 2 provides the drainage area, number of stations and abbreviation used for each sub-basin. For more information about the selected stations, please see Tables S1 to S4 in the Supplement. At each stream, we convert the daily discharge data into runoff depth in millimeter per week by dividing the weekly mean discharge to contributing basin area to obtain a set of comparable streamflow series (Whitfield and Cannon, 2000; Déry et al., 2009). Figure 4 shows the

distributions of the considered 106 RHBN stations over major drainage basins and sub-basins.

**Table 2.** Main sub-basins of the four Canadian major drainage basins along with their drainage areas, abbreviations and the number of RHBN stations within their territory used in this study.

| Major Basin | Sub-basin | Area (1000 km$^2$) | # of stations | Abbreviation |
|---|---|---|---|---|
| Pacific | Yukon | 330.4 | 4 | P1 |
| | Seaboard | 334.2 | 8 | P2 |
| | Fraser | 232.5 | 8 | P3 |
| | Columbia | 102.8 | 10 | P4 |
| Atlantic | Seaboard | 499.7 | 28 | At1 |
| | St. Lawrence | 860.1 | 16 | At2 |
| | Saint John- St. Croix | 41.9 | 6 | At3 |
| Arctic | Seaboard | 1,739.3 | 2 | Ar1 |
| | Lower Mackenzie | 1,321.1 | 7 | Ar2 |
| | Peace Athabasca | 482.7 | 3 | Ar3 |
| Hudson Bay | Western & Northern HB | 1,243.9 | 3 | H1 |
| | Northern Quebec & Ontario | 1,889.2 | 3 | H2 |
| | Nelson | 1,138.5 | 8 | H3 |



**Figure 4.** The distribution of the selected 106 RHBN streamflow stations within the major Canadian drainage basins and sub-basins.

## 4 Results

We apply the proposed framework to 106 selected RHBN streams using the thirty streamflow features introduced in Table 1. Note that in this section we only consider a decadal timeframe to calculate the streamflow features, cluster centers, and membership values. We repeat the algorithm with 15- and 20-year timeframes and compare the findings with those presented here in Sect. 5.2 to address the sensitivity of our results to the length of timeframe.

### 4.1 Identifying natural streamflow regimes in Canada

By having a decadal timeframe, we consider the period of 1966-1975 as the baseline. The optimal number of clusters is identified empirically from the pool of $c = \{2, 3, …, 10\}$, using the three validity indices introduced in Section 2.3. Figure 5





summarizes the results, showing $c = 6$ as the optimal number of clusters. Table 3 introduces these six regimes along with their notation and architype streams. Architype streams are those streams that have the highest association to the identified regime types and can represent the characteristics of a given regime better than other members of the cluster.

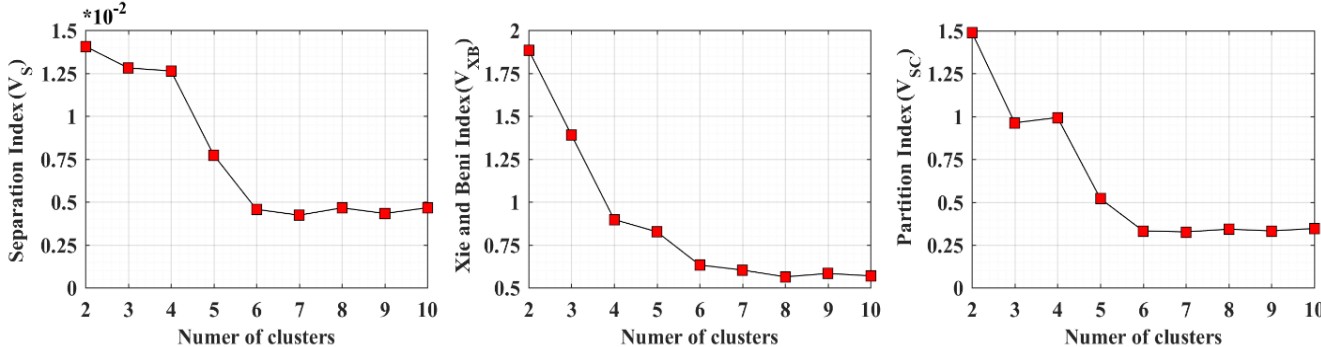

**Figure 5.** Variation in the Separation, Xie and Beni and Partition indices by altering the numbers of clusters.

**Table 3.** Six identified regime clusters along with their labelled regime type and architype stream.

| Cluster | Regime type | Architype (representative) stream |
|---|---|---|
| C1 | Glacial | Kazan River above Kazan Falls (HYDAT ID: 06LC001) |
| C2 | Nivo-glacial | Clearwater River near Clearwater Station (HYDAT ID: 08LA001) |
| C3 | Nival | Matawin River at Saint-Michel-des-Saints (HYDAT ID: 02NF003) |
| C4 | Nivo-pluvial | Gander River at Big Chute (HYDAT ID: 02YQ001) |
| C5 | Pluvio-nival | Beaver Bank River near Kinsac (HYDAT ID: 01DG003) |
| C6 | Pluvial | Sproat River near Alberni (HYDAT ID: 08HB008) |

Figure 6 shows the distribution of streams belong to each flow regimes across Canada, where red stars represent the architype stations for each regime type. The larger the size of circles are, the higher the degrees of membership are to each cluster. Note that for each regime type, we only consider/show streams with membership values of 0.1 and higher to eliminate streams with insignificant memberships and be able to provide a synoptic look at distribution and physical characteristics of streams within each cluster. As Fig. 6 shows the identified clusters are geographically concentrated and referring to the already-known regimes across the country (see Whitfield, 2001; Bawden et al., 2015; Burn and Whitfield, 2016; Najafi et al., 2017; Bush and Lemmen, 2019).

The first cluster (C1) resembles glacial flow regimes, characterized by gradual rise after spring snowmelt, prolonged peak discharge throughout summer, gradual recession during fall, and low runoff in winter (Déry et al., 2009). Glacial regime spreads mostly over Pacific (34%), Arctic (32%) and Hudson Bay (32%). The Kazan River releasing into the Baker Lake in Nunavut is the architype stream for this regime type.







**Figure 6.** The distribution of the identified regime types across Canada's major basins and sub-basins. The size of circles is proportional to the membership degrees. Only streamflow stations with degrees of membership of 0.1 or higher are shown in each panel. The red stars are the architype stations related to each regime type.





The second cluster (C2) resembles nivo-glacial streamflow regime, dominated by combined melts of accumulated snow and glacier storage (Eaton and Moore, 2010). The hydrograph shape is similar to glacial regime, but with an earlier and sharper recession. Peak flow usually occurs in June and July and the glacier melt maintain flows during late summer and early fall (Moore et al., 2012; Schnorbus et al., 2014). Nearly 46% of nivo-glacial streams are located in Pacific, especially in Fraser and Columbia. The rest are distributed in Hudson Bay (nearly 22%), particularly around the Rocky Mountains, Arctic (19%)

and Atlantic (13%). The Clearwater River near Clearwater in southern Alberta is the representative stream of nivo-glacial regime among the considered RHBN stations.

        The third cluster (C3) features nival regime, identified by short high flow period in early spring, sharp recession in summer and high flow variability throughout a typical year. 90% of nival streams are located in Atlantic, particularly around southwestern part of St. Lawrence. The rest of streams with nival regime located in southern part of Pacific and northern

Ontario. The Matawin River originated from the lake Matawin in Quebec is the architype for nival regime.

        The fourth cluster (C4) represents nivo-pluvial regime, with two distinct peaks, one in fall due to high precipitation, and one in spring induced by snowmelt. In this regime, snowmelt is the main contributor to runoff; and therefore, the maximum annual discharge is in the spring. This regime has a low flow in early fall and late winter (Hock et al., 2005). 80% of streams with nivo-pluvial regime are located in the southern Atlantic, with few streams in the southern Pacific, and in Hudson Bay.

Gander River at Partridgeberry Hill in eastern Newfoundland is the architype for this regime.

        The fifth cluster (C5) resembles the hybrid pluvio-nival regime, in which annual stream is influenced more by rainfall around later fall, followed by light increase in discharge due to snowmelt in early spring (Kang et al., 2016). The majority of streams with pluvio-nival regime (81%) are located in lower parts of Atlantic Seaboard. The other 19% of streams are located in Pacific, mainly in southern Fraser and Pacific Seaboard. Beaver Bank River in Nova Scotia is the representative stream for

this regime type.

        Finally, the sixth cluster (C6) features the pluvial regime, in which runoff is dominated by heavy precipitation, especially during winter and lower runoff during summer (Wade et al., 2001; Whitfield, 2001). Due to variations in rainfall, there is a high variability in winter flows. Pluvial regime is only seen in the Vancouver Island with the Sproat River near Alberni being the architype stream.

**4.2 Detection of change in streamflow regimes**

To highlight the changes occurred in each regime during the study period, we first look at changes in the shapes of annual streamflow hydrographs in the architype streams, revealed in two distinct decadal episodes during the first and the last decadal timeframes (i.e., 1966 to 1975 *vs.* 2001 to 2010). Figure 7 illustrates the results, in which expected annual hydrographs with weekly time steps are shown in solid black and red lines for 1966 to 1975 and 2001 to 2010, respectively. The grey and the

pink envelopes illustrate the variability in the annual hydrograph during the first and the last decadal periods. Although annual



hydrograph has altered in all architype streams (see the change in membership degrees in parentheses), forms of change in each regime type can be quite different.

**Figure 7.** Alterations in the decadal streamflow regimes at the archetype streams through time. The envelopes of 10-year annual hydrographs for the earliest (1966 to 1975) and the latest (2001 to 2010) episodes are shown with grey and pink colors, respectively. The expected annual hydrograph during the earliest and the latest 10-year periods are shown in solid black and red lines. The change in the membership degree of each archetype stream is shown within parentheses.





To have a better look at shifts in streamflow regime throughout the country, we calculate the decadal membership values in all streams and throughout the study period and apply the Mann-Kendall trend test with the Sen's Slope on the time series of decadal memberships. Figure 8 summarizes the results. In this figure, rows and columns are related to sub-basins (see Table 2) and regime types (see Table 3), respectively. In each panel, the *x*-axis shows the decadal timeframes from the beginning of the data period (i.e. 1966 to 1975 being the 1st timeframe and 2001 to 2010 being the 35th timeframe). The *y*-axis shows and

natural streams in each sub-basin sorted north to south from top to bottom. Grey bars represent the time series of decadal memberships and color bars show the associated trends in the anomalies of membership values. While blue and red colors show decreasing and increasing trends, the intensity of colors shows significant and insignificant trends (*p*-value $\leq$ 0.05). Note that similar to Fig. 6, we only show membership degrees of 0.1 and higher.

   Despite being the smallest among the four major drainage basins in Canada, the Pacific shows the largest diversity in

streamflow regimes and includes all six regime types identified (see panels in the first four rows of Fig. 8). During the data period considered, patterns of change in regime types are subject to significant variation across the Pacific. For instance, considering the glacial regime, the belongingness is significantly increasing in Yukon (streams in the very top left panel, identified with P1/glacial); however, it is gradually diminishing in the northern Pacific Seaboard over the considered timeframe (streams in the panel below, identified with P2/glacial). In Fraser (panel P3/glacial), a mixed pattern of change is observed

with only one stream showing significant strengthening of glacial regime. In Columbia (panel P4/glacial), patterns of change are different between the northern (increasing trends) and southern parts (decreasing trends). Such sub-regional differences in the form of change are quite common. For instance, while in Columbia the degrees of membership to nivo-glacial regime mainly decrease, mixed patterns of change are observed in the Fraser.

   The same analysis can be applied to other basins. After the Pacific, the Atlantic basin shows the highest diversity in the

streamflow regime by having five regime types within its territory. The dominant streamflow regimes in the Atlantic basin are nival and nivo-pluvial regimes, where 46 and 41 streams out of the total of 50 natural streams considered in this basin show some levels of belongingness to these two regime types, respectively. From these 46 and 41 streams, 29 and 19 streams show increasing trends in belongingness to nival and nivo-pluvial regimes, respectively. The mixed pattern of change is subject to large geographic variations at the sub-basin scales. For instance in the Saint John- St. Croix, the associations to nivo-pluvial

regime become stronger during the study period. Similarly, half of the 50 considered streams in Atlantic are associated to some extent with pluvio-nival regime. These streams also show a mixed pattern of change in their belongingness to this regime, similar to the glacial and nivo-glacial regimes that are less common in the Atlantic basin.

   In contrast to the Pacific and the Atlantic, the Arctic has the least diversity in the streamflow regime. All considered 12 streams are associated with large degrees to glacial regime, out of which five and six streams show increasing and decreasing

trends in the membership, respectively. In addition, nine out of 12 streams located in the Arctic are associated with nivo-glacial regime, for which the degrees of membership increase in six stations. There is only one stream in the Peace Athabasca, showing an increase in belongingness to nivo-pluvial regime during the study period.





**Figure 8.** The evolution in the degrees of membership to each regime type in 106 considered RHBN streams along with the corresponding
Sen's slope. For each stream, the shades of grey show decadal memberships over the period of 1966 to 2010. The color bar shows the
direction and significance of the Sen's slope of the trend in the anomalies of memberships. Positive and negative trends are shown with red
and blue colors, respectively. Sharper colors show significant cases.





In the Hudson Bay basin, 12 and 11 streams out of 14 streams show some levels of association with glacial and nivo-glacial regimes, out of which nine and seven streams show departure from these regimes during the study period, respectively. Six

out of 14 considered streams in this basin are associated with some degrees to nival and nivo-pluvial regimes. These streams are located mainly in the Western & Northern Hudson Bay and Nelson sub-basins. The membership values for both regime types show increasing trends, in which three streams exhibit significant trends.

## 4.3 Identifying forms of transformation in streamflow regimes

As noted in Sect. 2.4, representing streamflow regimes using fuzzy clusters implies that the sum of memberships to all regimes

at each stream and during each timeframe equals to one – see Eq. 2(b). As a result, the existence of trend in membership values can be taken as a direct evidence for transformation of flow regime from one type to others. This sets the scene for investigating regime shifts quantitatively, as the sign and the slope of change in membership degrees can provide a notion for the direction and intensity of the shifts between regime types. Figure 9 summarizes the results of this analysis for the time series of decadal memberships shown in Fig. 8 in grey. Rows display the considered 106 RHBN stations, categorized by their corresponding

basins and sub-basins (see Table 2). The streams in each sub-basin are sorted north to south from top to bottom, similar to Fig. 8. Columns are related to the six regime types (see Table 3). Similar categorization is made on the x-axes in all panels. Considering each panel, the regime type identified by the column filled with diagonal lines (i.e., identical regime types) is the potential receiver regime that the increase in its membership should coincide with the decrease in the membership values of dissimilar regime types (i.e., givers), shown in the x-axes. Considering each pairs of dissimilar regime types, shades of grey

represents dissimilar directions of change, quantified according to the sign of Kendall's tau. Note that we only display those pairs of flow regimes, in which evolutions in membership degrees are significantly dependent according to Kendall's tau ($p$-value ≤ 0.05) for streams with membership degrees of 0.1 and higher. In each panel, cells related to identical regime types are shaded in diagonal lines.

In the Pacific, the dominant regime shift is observed between glacial and nival regime (in 13 streams; panels of P1 to P4

under glacial and nival). At the sub-basin scale, however, the dominant regime shift may be different. In Yukon, gradual shifts are observed from nival to glacial regime (panel P1/glacial); whereas, in the northern Pacific Seaboard, one stream depart from glacial and shift toward nivo-glacial (panel P2/nivo-glacial). The other stream in this region mainly shift from pluvio-nival to nivo-glacial regimes. In the southern part of Pacific Seaboard, a mixed pattern of transition happens mainly between pluvio-nival and pluvial regime. In Fraser, a mixed pattern of transition between glacial and nivo-glacial happens in the northern part

(panel P3/glacial and nivo-glacial); whereas, one stream in the southern part shift from nivo-glacial to pluvio-nival regime (panel P3/pluvio-nival). In Columbia, two streams located in the northern part shift from nivo-glacial to glacial regimes (panel P4/glacial); while, streams located in the southern part show a strong tendency to depart from glacial to nival as well as from glacial to nivo-pluvial (panels P4/nival and nivo-pluvial). Additionally, a coherent pattern of shift from nivo-glacial to nival as well as from nivo-glacial to nivo-pluvial is observed for streams located in Columbia (panels P4/nival and nivo-pluvial).

This has led to disappearance of glacial and nivo-glacial regimes in some of the streams in this region – see Fig. 8.





**Figure 9.** Mapping shifts in natural streamflow throughout Canada during 1966 to 2010. Rates of shift among various regime types in each stream are shown by shades of grey that quantifies how much decline in the giver regimes shown in the x-axes in each panel can result into incline in the receiver regime type corresponding with the column in which the panel is located. Columns filled with diagonal lines show the identical regime types with the receiving regimes identified in the column where the panel is located.





In the Atlantic, dominant transition is between nivo-pluvial and pluvio-nival (in 27 streams; panels of At1 to At3 under nivo-pluvial and pluvio-nival), as well as between nival and pluvio-nival regimes (in 25 streams; panels of At1 to At3 under nival and pluvio-nival). At the sub-basin scale, however, the pattern of transition between flow regimes is not homogenous. In

the northern part of Atlantic Seaboard, a coherent pattern of shift from pluvio-nival to nival regime is observed for five out of 10 streams (panel At1/nival); whereas, four streams in this region show an intense shift mainly from pluvio-nival to nivo-pluvial regime (panel At1/nivo-pluvial). Similar to the streams in the northern part of Atlantic Seaboard, nine streams located in the southern part shift from pluvio-nival to nivo-pluvial. Moreover, 12 streams in this regions shift from pluvio-nival to nival regime. There are also three stations gradually shift from nival to pluvio-nival regime in this region. In the northern part

of St. Lawrence, a shift mainly from nival to glacial is seen in two streams. Reversely, in the southern part seven streams shift mainly from glacial to nival regime (panel At2/nival). Additionally, six streams show tendency to shift mainly from glacial to nivo-pluvial regime. In Saint John- St. Croix, similar to the streams located in Atlantic Seaboard, five streams shift from pluvio-nival to nivo-pluvial regime and the rates of transition in northern streams are more pronounced than the southern parts (panel At3/nivo-pluvial).

In the Arctic, there are frequent transformations between glacial and nivo-glacial regimes (eight out of 12 streams; panels of Ar1 to Ar3 under glacial and nivo-glacial regimes). Similar pattern of shift is observed at the sub-basin scale. In the Seaboard and Lower Mackenzie, the dominant form of transition is from glacial to nivo-glacial regime (see panels of Ar1 and Ar2 under glacial and nivo-glacial regimes). In the Peace Athabasca, two streams shift from nivo-glacial to glacial, while the other slightly departs from glacial into nivo-pluvial regime (panel Ar3/nivo-pluvial).

In the Hudson Bay, the dominant transitions are between glacial and nival regimes (seven out of 14 streams; panels H1 to H3 under glacial and nival regimes), as well as between nivo-glacial and nival regimes (seven out of 14 streams; panels H1 to H3 under nivo-glacial and nival regimes). This is consistent with the pattern of shift at the sub-basin scale. In the Western & Northern Hudson Bay, there are obvious shifts mainly from glacial to nival as well as from nivo-glacial to nival (panel H1/nival). Additionally in this region, a shift from glacial to nivo-pluvial as well as from nivo-glacial to nivo-pluvial is

observed. In Northern Quebec & Ontario, natural streamflow regime shifts from glacial toward nivo-glacial regime (panel H2/nivo-glaical). In the northern part of Nelson, three streams shift mainly from glacial to nival regime (panel H3/nival). Unlike the northern part, one stream in the southern part shifts from nivo-glacial to glacial regime. The other stream also show a tendency to shift from nival to glacial regime.

### 4.4 Attribution of regime shift to changes in streamflow characteristics

The methodology presented in Sect. 2.5 provides an empirical basis to evaluate the association of the regime shifts to changes in streamflow characteristics using the coefficient of determination. Figures 10 and 11 below show this analysis between decadal memberships and decadal means and variances of the 15 IHAs, respectively. In both figures, each panel corresponds with a sub-basin and a regime type, arranged in rows and columns, respectively. In each panel, streamflow features are





displayed in the x-axis, i.e., the means of the 15 IHAs are displayed in Fig. 10 and variances are shown in Fig. 11. y-axes show
the streams at each basin, sorted from north (top) to south (bottom) of each panel. In both figures, we only show and discuss
those streamflow characteristics that have significant dependencies with variations in membership degrees based on the
Kendall's tau ($p$-value ≤ 0.05) for streams with membership degrees of 0.1 and higher. Kendall's tau also identifies the sign
of dependence. The blue and red colors show positive and negative dependencies and the color saturation is based on the
coefficient of determination.

In Yukon, the shift from nival to glacial regime is attributed to the earlier timing of the annual low flow, earlier timing and
higher variability in the annual high flow, as well as increasing September flow along with increasing variability in April/May
flows (see panel P1 under glacial regimes in Figs. 10 and 11). In northern Pacific Seaboard, increasing winter flows and
monthly flow in May, earlier timing of low flow along with increasing variability in March, May and yearly flows are
corresponding with the shift from glacial into nivo-glacial regime (see panels P2 under glacial and nivo-glacial regimes in
Figs. 10 and 11). In the southern parts of Pacific Seaboard, however, decreasing mean and variance of the annual flow,
decreasing mean flows in July and September, decreasing mean and variance of seasonal flow in fall along with declining
variability in monthly February flow resulted into the shift from pluvial into more pluvio-nival regime. Reversely, increasing
mean and variance of the annual and monthly flow in July, increasing mean and variance of monthly October flow, increasing
monthly flow in January, as well as increasing variation of winter flow correspond with shift from pluvio-nival to pluvial
regime. In the northern part of Fraser, the shift from glacial to nivo-glacial is attributed to increasing mean and variance in
annual and summer flows, as well as monthly flows in May and June along with increasing variation in timing of low flow
and spring flows. The opposite shift from nivo-glacial to glacial, however, corresponds to decreasing mean and variance in
annual flow, decreasing monthly flow in July and October, earlier timing of high flow and decreasing variation of monthly
flows in May, August and September (see panels of P3 under glacial and nivo-glacial regimes in Figs. 10 and 11). In southern
part of Fraser, the shift from nivo-glacial to pluvio-nival is attributed to decreasing summer flows, earlier timing of high flows,
increasing mean and variability in November and April flows. In northern Columbia, the shift from nivo-glacial to glacial
regime is attributed to decreasing annual and summer flows along with decreasing variability in annual and monthly flows
during August and September (see panels P4 under glacial and nivo-glacial regimes in Figs. 10 and 11). In the southern parts
of Columbia delayed timing and higher variability of the annual low flow, along with an earlier timing of the annual high flow,
increasing mean flows in April and November, decreasing flow in September and increasing variance of November's flow
lead to the shift from glacial to nival regime (see panels under P4/glacial in Figs. 10 and 11). The changes in the aforementioned
characteristics also lead to shift from nivo-glacial to nival regime in southern Columbia.

In the northern part of the Atlantic Seaboard, increasing monthly flow in April, decreasing monthly flow during June,
delayed and less variable timing of annual low flows, less variation of annual timing of high flows as well as decreasing mean
and variation of August's monthly flow correspond with the shift from pluvio-nival to nival regime (panels under At1/pluvio-
nival and nival in Figs. 10 and 11). Similarly in the southern part of Atlantic Seaboard, the shift from pluvio-nival to nivo-
pluvial is attributed to decreasing annual and monthly flow in May, June and August, increasing March monthly flow along





with decreasing variation of monthly flow in February. The changes in the above-mentioned characteristics also lead to shift from pluvio-nival to nival regime. In few stations, however, the shift from nival to pluvio-nival regime corresponds with

decreasing monthly flow in January, February, May, June and later timing of low flow. In northern St. Lawrence, the shifts from nival to glacial in two streams are attributed to decreasing annual, winter, summer and monthly flows in June as well as less variation in monthly flows in February, May, June and timing of low flow (see panels under At2/glacial in Figs. 10 and 11). In southern part of St. Lawrence, the shift from glacial to nival regime is attributed to increasing (decreasing) mean and variation in monthly flows in May (September), decreasing flow in October, increasing February  flow as well as increasing

variance in timing of low and January's monthly flows (see panels At2 under glacial and nival regimes in Figs. 10 and 11). In Saint John- St. Croix, decreasing annual and monthly flows in February, May and June along with decreasing expected value and variability in October and August correspond to the shift from pluvio-nival to nivo-pluvial regime (panels under At3/pluvio-nival in Figs. 10 and 11).

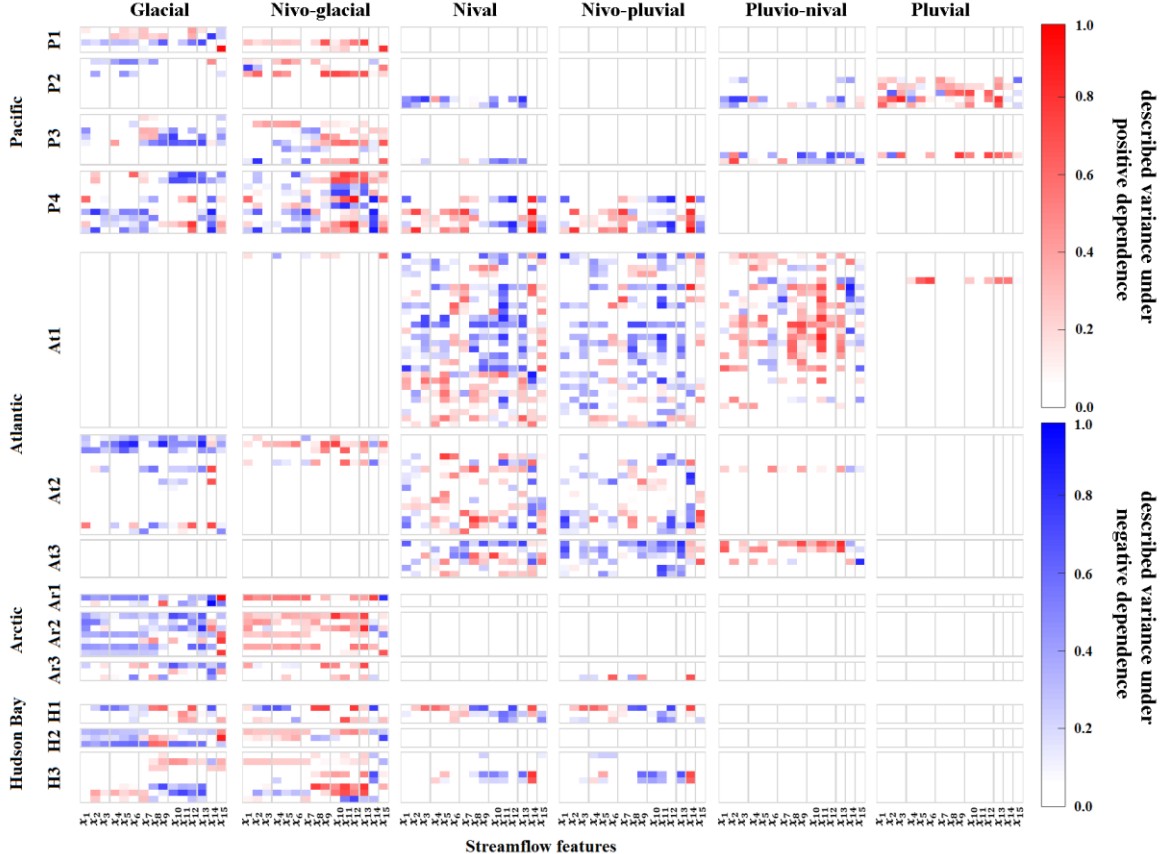

**Figure 10.** The alterations in regime types for 106 RHBN streams attributed to the first moments of the 15 IHA considered. Shades of red and blue show the positive and negative dependencies between changes in streamflow features and the degrees of membership, respectively. Color saturation shows the coefficient of determination between changes in the streamflow features and the degrees of membership representing the percentage of described variance in changes of streamflow regime by changes in streamflow features.

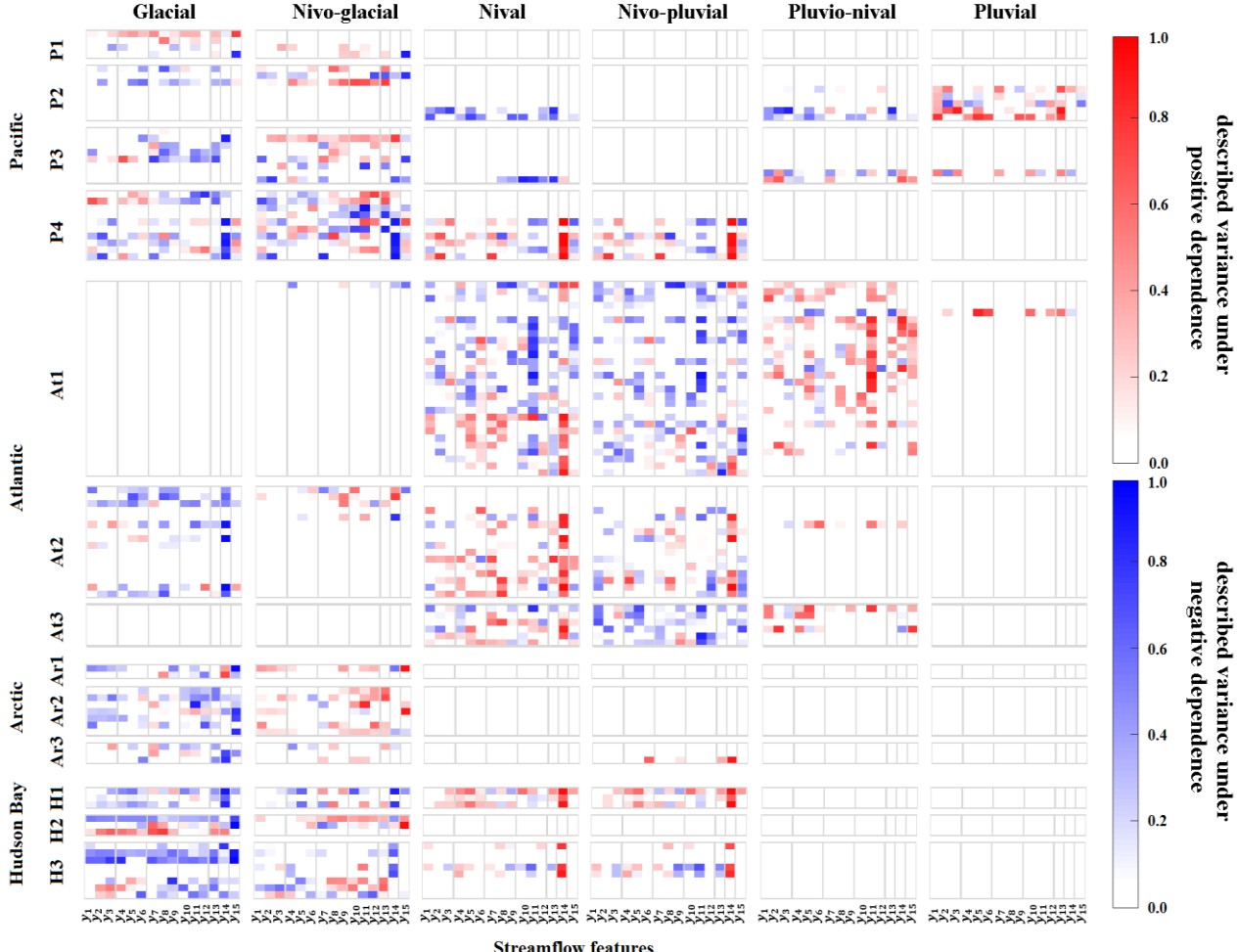

**Figure 11.** The alterations in regime types for 106 RHBN streams attributed to the second moments of the 15 IHA considered. Shades of red and blue show the positive and negative dependencies between changes in streamflow features and the degrees of membership, respectively. Color saturation shows the coefficient of determination between changes in the streamflow features and the degrees of membership representing the percentage of described variance in changes of streamflow regime by changes in streamflow features.

In the Arctic Seaboard, earlier but more variable timing of annual high flow, increasing mean and variability of seasonal flow in fall, along with increasing winter flows and heightened variability in monthly flow in June collectively correspond with the regime shift from glacial into nivo-glacial regime (see panels Ar1/glacial in Figs. 10 and 11). In Lower Mackenzie, increasing annual and seasonal flows during fall and winter, increasing June's monthly flow, along with heightening variability in the timing of high flows correspond with a gradual shift from glacial to nivo-glacial regime (see panels under Ar2/glacial in Figs. 10 and 11). Having said that, some other streams in this region have experienced a reverse shift from nivo-glacial to





glacial regime, due to decreasing monthly flows in October and August as well as decreasing annual flow. In Peace Athabasca, the reverse shift from nivo-glacial to glacial regime corresponds with the earlier and less variable timing of low flows as well as decreasing monthly flow in July (see panels under Ar3/nivo-glacial in Figs. 10 and 11). The other stream in this region shifts
from glacial to nivo-pluvial regimes, due to increasing monthly flow in June, decreasing monthly flow in May, increasing mean and variation of monthly flow in March along with delayed and more variation in timing of low flow.

In Western & Northern Hudson Bay, the delayed and more variable timing of annual low flow, along with increasing monthly flows in December, January and February, decreasing monthly flows in May, June, and September, and increasing variability in February's monthly flow lead to the regime shift mainly from glacial to nival regime. Same set of characteristics
also led to regime shift from nivo-glacial to nival regimes (see panels of H1 under nivo-glacial and nival regimes in Figs. 10 and 11). In Northern Quebec & Ontario, the shift from glacial into nivo-glacial regime corresponds to increases in annual and seasonal flows in fall, winter and summer, and a decreased and less variable monthly flows in May, along with a decreased flow in June and an increased variability in timing of high flows (see panels H2/glacial in Figs. 10 and 11). In the northern parts of Nelson, the shift from glacial to nival regime is attributed to decreasing monthly flow in May and June, decreasing
seasonal flow in summer along with increasing variability in timing of annual low and high flows, annual average flow as well as seasonal flows during fall, spring, winter and summer (see panel H3/glacial in Figs. 10 and 11).

# 5 Discussion

## 5.1 Summary of findings and positioning against earlier studies

By having four ocean-drained basins, spread over more than 6% of the global land area, Canada exhibits a large diversity in
its natural streamflow regime. Canadian rivers, rolling coast to coast to coast, support important socio-economic activities such as agricultural and hydropower production, and feed some of the world's most important freshwater bodies that are home to various (some endangered) wildlife species. Natural streamflow regime in Canada, however, is going through drastic changes in recent years. The literature of Canadian hydrology is rich in terms of documenting changes in streamflow characteristics across the country. There are a large body of studies, reporting shift in streamflow regime across different regions due to
changes in temperature patterns, magnitude and form of precipitation, snowmelt and snow accumulation processes as well as glacier retreat and permafrost degradation. Thanks to pioneering works of so many before us, including the iconic late northern hydrologist, Richard Janowicz, to whom this paper is dedicated. Having said that, to the best of our knowledge, our work is the first study in which a fully algorithmic framework is used to provide a temporally-homogeneous pan-Canadian view on the recent shifts in natural streamflow regime across the country.
Our results presented in Sect. 4.4 reveal that shifts in streamflow regimes can be attributed to simultaneous changes in a large ensemble of streamflow characteristics. This conclusion is consistent with the earlier findings on changes in natural streamflow characteristics across most Canadian basins and sub-basins; yet our study reveals some new changes in streamflow





characteristics that have been previously overlooked or remained unknown. To better position our results against earlier studies, Table 4 summarizes our findings in terms of dominant regime shifts and associated changes in streamflow

characteristics at the sub-basin scale. Note that in the majority of sub-basins, there is an obvious divide between dominant regime shifts in northern and southern regions; and therefore, they are separated from one another. Table 4 also makes a clear distinctions between the earlier findings that reconfirmed in the current study, and those exclusively found in our work.

Two important findings can be obtained from this comparison. First and foremost, our study provides new sets of understanding on shift in streamflow regime and forms of alteration in streamflow characteristics in two regions in Canada

that have not been previously studied, i.e. northern Fraser and southern St. Lawrence. In both regions, shifts in streamflow regime is attributed to changes in multiple streamflow characteristics. Second, earlier studies often looked only at the changes in expected monthly, seasonal and annual flows. In fact, evaluating changes in variability in streamflow characteristics remained mainly limited to timing of low and high flows. Our study clearly shows that changes in variability of monthly, seasonal and annual flows can be important drivers of shift in streamflow regime across majority of sub-basins in Canada. This

is another line of evidence for the complex and multifaceted nature of change in streamflow regime, and the need for simultaneous look at alterations in both expected values and variability of streamflow characteristics to diagnose changes in natural streamflow regime. It should be also noted that earlier studies may have different study periods, and may include streams that are not particularly within the RHBN streams.

**5.2 Addressing uncertainty**

The results provided in Sect. 4 are based on decadal timeframes. To address the uncertainty in our findings due to this assumption, we repeat implementing the proposed algorithm with 15- and 20-year timeframes, and look at the differences in our findings presented in Sect. 4.1 to 4.4.

In terms of clustering, our results show that the number of optimal clusters does not change by altering the timeframe's length. In addition, as shown in Fig. S1 in the Supplement, there are not significant changes in the cluster centers. This

highlights the robustness of the FCM in identifying distinct flow regimes.

We also look at possible differences in the direction of trends in membership degrees, presented in Sect. 4.2 (Fig. 8). The corresponding results obtained by 15- and 20-year timeframes are presented in Figs. S2 and S3 in the Supplement. Figure 12 (left column) intercompares the results obtained by 10-, 15- and 20-year timeframes in terms of percentages of similarities in the direction of trends during 1966 to 2010 at each basin. In brief, there are at least 80% agreements between the results

obtained in the Pacific and the Arctic basins. Relatively, there are more discrepancies in the direction of trends identified by different timeframes in the Atlantic and Hudson Bay basins. This is particularly the case for glacial regime types in the Hudson Bay and for nival and nivo-pluvial in the Atlantic, for which the results are less consistent among different timeframes; yet, in the case with largest disagreement between the results obtained by different timeframes (i.e. nivo-pluvial regime in Atlantic), there is more than 60% agreement between the trend results obtained by 10-, 15- and 20-year timeframes.





**Table 4.** Positioning the finding of the current study against to earlier studies on changes in streamflow characteristics across major Canadian basins and sub-basins

| Basin | Sub-basin (stream location) | Dominant regime shift | Earlier findings on changes in streamflow characteristics (reconfirmed in current study) | New findings on changes in streamflow characteristics (discovered exclusively in current study) |
|---|---|---|---|---|
| Pacific | Yukon | Nival to glacial | Earlier timing of low and high flows; higher variability in timing of high flows (Burn 2008; Brabets and Walvoord, 2009; St. Jacques and Sauchyn, 2009) | Increasing flow in September; increasing flow variability in April and May |
| | Seaboard (north) | Glacial to nivo-glacial | Increasing winter flows (Déry et al., 2009) | Increasing monthly flow in May; earlier timing of low flow; increasing variability in March, May and annual flows |
| | Seaboard (south) | Glacial to nival | Decreasing annual and monthly flow from April to June; decreasing flow in fall (Déry et al., 2009; Pike et al., 2010) | Delayed and more variable timing of annual low flow; increasing variability in February's monthly flow |
| | Fraser (north) | Case 1: glacial to nivo-glacial; Case 2: nivo-glacial to glacial | No earlier study in this region was found. | Case 1: Increasing mean and variance in annual and summer flows; increasing monthly flows in May and June; increasing variation in timing of low flow and the quantity of spring flows. Case 2: Decreasing mean and variance of annual flow; decreasing monthly flows in July and October; earlier timing of high flow; decreasing variability of monthly flows in May, August, September |
| | Fraser (south) | Nivo-glacial to pluvio-nival | Decreasing summer flows (Stahl and Moore, 2006); Increasing variability in monthly flows in November and April (Déry et al., 2012; Thorne and Woo, 2011) | Earlier timing of high flows; increasing mean monthly flows in November and April |
| | Columbia (north) | Nivo-glacial to glacial | Decreasing annual and summer flows (Stahl and Moore, 2006; Fleming and Weber, 2012; Forbes et al., 2019) | Decreasing variability in annual flow, and monthly flows of August and September |
| | Columbia (south) | Glacial to nival | Increasing flow in April and decreasing flow in September (Whitfield and Cannon; 2000; Whitfield, 2001); Earlier timing of high flow (Burn and Whitfield, 2016; Burn et al., 2016) | Delayed timing and higher variability of the annual low flow; increasing mean and variance of flow in November's flow |
| Atlantic | Seaboard (north) | Pluvio-nival to nival | increasing spring flows, corresponding to increased snow precipitation (Thistle and Caissie, 2013) | Increasing monthly flow in April; decreasing monthly flow in June; delayed and less variable timing of low flows; less variation in annual timing of high flows; decreasing mean and variation of monthly flow in August |
| | Seaboard (south) | Case 1: pluvio-nival to nivo-pluvial; Case 2: nival to pluvio-nival | Case 1: decline in the annual flow (Whitfield and Cannon, 2000; Yue et al., 2003; Thistle and Caissie (2013) Case 2: decline in winter flows, probably due to positive AMO (Whitfield and Cannon, 2000; Assani et al., 2012) | Case 1: Decreasing monthly flow in May, June and August; increasing monhly flow in March; Decreasing variability in February's monthly flow Case 2: Decreasing monthly flow in May and June; later timing of low flows |
| | St. Lawrence (north) | Nival to glacial | lower variations in timing of low flow (Thistle and Caissie, 2013) | Decreasing annual flow as well as seasonal flows in summer and winter; decreasing monthly flows in June, less variation in monthly flows of February, May, June |
| | St. Lawrence (south) | Glacial to nival | No earlier study in this region was found. | Increasing mean and variation in monthly May flows; decreasing mean and variation in September flows; decreasing flow in October, increasing flow in February; increasing variance in timing of low flows; increasing variability in January's monthly flows |
| | Saint John- St. Croix | Pluvio-nival to nivo-pluvial | Decreasing monthly flow in May (Kingston et al., 2011) | Decreasing annual flow; deceasing monthly flows in February and June; decreasing mean and variability of monthly flows in October and August |
| Arctic | Seaboard | Glacial to nivo-glacial | Earlier and more variable timing of high flows; increasing winter flows (Burn, 2008; Déry et al. 2016); earlier timing of high flows (Yang et al.; 2015) | increasing mean and variability of seasonal flow in fall, heightened variability in monthly flow in June |
| | Lower Mackenzie | Glacial to nivo-glacial | Increasing annual and winter flows (Smith et al., 2007; Walvoord and Striegl, 2007; St. Jacques and Sauchyn, 2009; Rood et al., 2016) | Increasing annual and seasonal flows during fall; increasing June's monthly flow; heightening variability in the timing of high flows |
| | Peace Athabasca | Nivo-glacial to glacial | Decreasing monthly flow in July (Yang et al., 2015) | earlier and less variable timing of low flows |
| Hudson Bay | Western & Northern Hudson Bay | Glacial to nival | Increasing winter flows; decreasing summer flows; increasing variability in winter flows (Déry et al., 2011, 2018) | Delayed and more variable timing of low flows; increasing variability in February's monthly flow |
| | Northern Quebec & Ontario | Glacial to nivo-glacial | Increasing annual and winter flows, increasing variability in timing of high flows | Increasing annual and seasonal fall and summer flows; decreasing and less variable monthly flows in May; decreasing monthly flow in June |
| | Nelson | Glacial to nival | Decreasing summer and fall flows Rood et al. (2008); Decreasing summer flows; increasing variability fall and spring flows (Déry et al., 2011) | Decreasing monthly flow in May and June; increasing variability of timing of low and high flows; increasing annual flow and seasonal flows in summer and winter |





The identification of direction of shift in streamflow regime (Sect. 4.3) is also performed with 15- and 20-year timeframes.
The results are presented in Figs. S4 and S5 in the Supplement and are intercompared with corresponding results obtained by

decadal timeframes in Fig. 12 (middle column). Our analysis shows that results obtained by 15- and 20-year timeframes are in
large agreements with the results obtained using decadal timeframes. Even for the case with the largest disagreement (i.e. nivo-
pluvial regime in the Atlantic), there is 86% agreement in terms of direction of shift in streamflow regimes, obtained by 10-
and 20-year timeframes.

In terms of attribution of regime shifts to changes in streamflow characteristics, again the results obtained by 15- and 20-

year timeframes (see Figs. S6 and S7 as well as S8 and S9 in the Supplement, respectively) show large agreement with the
results obtained by the decadal timeframe and presented in Sect. 4.4. According to the intercomparison made in Fig. 12 (right
column), there is at least 80% agreement in the results obtained by different timeframe lengths.

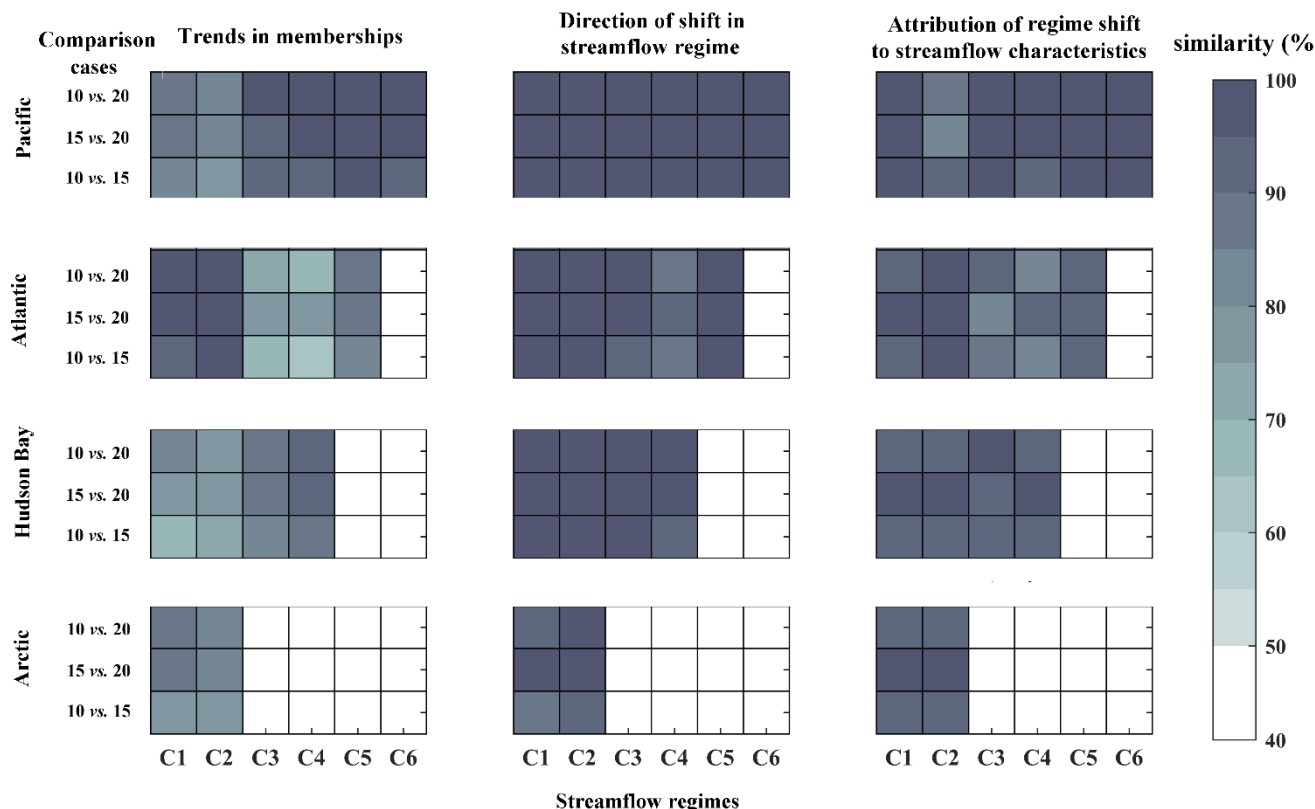

**Figure 12**. Similarities (in percentage) between the results obtained by 10-, 15- and 20-year timeframes related to trends in membership
values, direction of shift in streamflow regimes, and attribution of regime shift to streamflow characteristics in the four major Canadian river
basins.



## 6 Conclusions and outlook

This study presents an attempts toward a generic algorithmic framework for identifying changing streamflow regimes, not only in Canada but also globally. The proposed approach is based on two fundamental considerations. First, we recognize that streamflow regime is collectively formed by a large set of streamflow characteristics that can describe the expected annual streamflow hydrograph and the inter-annual variability around it. Second, we acknowledge that streamflow types are rather spectrums, not clear-cut states; and the transition from one regime type to another is gradual rather than abrupt. To accommodate these two considerations, we suggest representing streamflow regime types as intersecting fuzzy sets, in a way that the belongingness of each stream to each regime type can be quantified by a unique membership function. Accordingly, monitoring the trends in membership values in time and space can provide a basis to identify the regime shift from one type to another. In addition, analyzing the covariance of membership values with streamflow characteristics can provide a basis to attribute the regime shift to alterations in certain streamflow characteristics in time and/or space.

By applying the proposed procedure to 106 RHBN streamflow gauges in Canada, we provide a comprehensive look at forms and extents of change in natural streamflow regime during 1966 to 2010 throughout the country. We show that the considered natural streams in Canada can be categorized into six distinct regime types with clear physical and geographical interpretations. Analyses of trends in membership values show that alterations in natural streamflow regime are vibrant and can be different across major drainage basins in Canada. Overall, dominant modes of transition at the basin scale are between glacial and nival in the Pacific, between nivo-pluvial and pluvio-nival as well as nival and pluvio-nival in the Atlantic, between glacial and nivo-glacial in the Arctic, and between glacial and nival as well as nivo-glacial and nival regimes in the Hudson Bay. The details of change in streamflow regime, however, are subject to a large spatial variability within each drainage basin. For instance in the Pacific, the association to the glacial regime is increasing in Yukon and northern parts of Columbia and Fraser sub-basins; but it is significantly decreasing in the southern regions. This can be due to different manifestations of climate change, which are more revealed as temperature increases in the north – and therefore more glacial retreat – rather than growing ratios of rain over precipitation in the south, which shift the streamflow more toward rain-dominated regimes (Fleming and Clarke, 2003). Such north/south divides are observed in other drainage basins as well, e.g. in the Atlantic and between streams located in north and south of Bay of Fundy. Having said that, even within close proximity, e.g. between Yukon and northern Pacific Seaboard, differences in the evolution of streamflow regime can be significant. It has been pointed that this regional contrast can be due to existence of the larger glacier in Yukon, i.e., St. Elias Mountains, which exhibit a different response under the same climatic conditions (Fleming and Clarke, 2003; Stahl and Moore, 2006). This reconfirms the important role of landscape in regulating the streamflow response to climate change.

Climate-driven changes in the streamflow regime will have multiple impacts on socio-economic activities and ecosystem services in Canada and globally, and therefore should be managed with care. This requires understanding that water does not respect political boundaries. We strongly believe that Canada and the rest of the world need sustainable and well-integrated water management plans to face the challenges (and the opportunities) ahead of us during the current "*Anthropocene*".



*Data availability.* The analysis is based on data provided by the Reference Hydrometric Basins Network (RHBN) of Environment Canada. The dataset can be accessible through streamflow records of HYDAT, complied by Water Survey of Canada, http://www.ec.gc.ca/rhc-wsc/default.asp?lang=En&n=9018B5EC-1).


*Supplement.* The supplement related to this article is available online.

*Author contributions.* MZ, AN and SH designed the methodology; MZ, SH and JS developed the computational procedure; MZ and SH executed the numerical work and analyzed the results; MZ, AN developed manuscript outline and flow; MZ and
SH developed the artworks; MZ performed the literature review and wrote the first draft; AN, SH and JS commented and revised the paper; AN and MZ finalized the manuscript.

*Competing interests.* The authors declare that they have no conflict of interest.

*Acknowledgement and dedication.* Financial support for this study is provided by Canada Natural Science and Engineering Research Council through Discovery Grant Program, as well as Concordia University through various internal sources. We greatly acknowledge valuable inputs from Elmira Hassanzadeh of Polytechnique Montréal, on the earlier version of this paper. This study is conducted with love and sweat, and is dedicated to the memory of Richard Janowicz, the iconic northern hydrologist who made fundamental discoveries on recent changes in natural streamflow regime in the Great White North:
Canadian Hydrology will never forget you, Rick.

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
