# Peer review of "A global algorithm for identifying changing streamflow regimes: Application to Canadian natural streams (1966-2010)"

_Hydrology and Earth System Sciences, 2020_

## Referee Comment (RC1) · Anonymous Referee #1 · 30 Sep 2020

General comments

The analyses presented are interesting, and may be useful in establishing the changes in the regimes of Canadian rivers. Unfortunately, the results are undermined by their presentation. With some revisions, I believe that the paper can make a good contribution.

The writing needs quite a bit of editing. There are too many grammatical mistakes to list here, and the writing is often unclear.

There are missing articles in many of the sentences, such as the first one:

[Figure]

Page 1, Line 28

"Natural streamflow characteristics have been critical consideration"

This is sentence is missing the article "a" before "critical", or needs to make "consideration" a plural

In many sentences there are disagreements in number, i.e. between singular and plurals:

Page 2, Line 34

"some others determines"

As was stated above, the writing is often unclear, as in the caption of Figure 9:

"Figure 9. Mapping shifts in natural streamflow throughout Canada during 1966 to 2010. Rates of shift among various regime types in each stream are shown by shades of grey that quantifies how much decline in the giver regimes shown in the x-axes in each panel can result into incline in the receiver regime type corresponding with the column in which the panel is located. Columns filled with diagonal lines show the identical regime types with the receiving regimes identified in the column where the panel is located."

Specific comments

There appears to be only one gauging station within the Canadian Prairies. This is disappointing as the hydrology of the region is very important and has seen many effects of changes in climate. There are several RHBN stations within the prairies, according to this website https://www.canada.ca/en/environment-climate-change/services/water-overview/quantity/monitoring/survey/data-products-services/reference-hydrometric-basin-network.html.

Whitfield et al. (2020) grouped responses of streams into 3 clusters in the Prairies and adjacent areas, using a very different clustering methodology. I assume that there were

no other prairie stations which met the authors' criteria. However, it would be good to have this explained. Would the use of a slightly different analysis period have allowed the inclusion of more prairie streams?

Whitfield, P.H., Shook, K.R., Pomeroy, J.W., 2020. Spatial patterns of temporal changes in Canadian Prairie streamflow using an alternative trend assessment approach. Journal of Hydrology 582, 124541. https://doi.org/10.1016/j.jhydrol.2020.124541

Although hydrologists are used to working with river basins, grouping the stations by basin is not always useful. As shown in Table 2, Canadian river basins are very large. Wong et al. (2017) identify 15 ecozones in Canada, many of which are spanned by single basins. For example, the Nelson River system spans the Montane Cordillera, Prairies, Boreal Plains, Canadian Shield and the Hudson Plain. Stations in differing ecozones would not be expected to behave in similar ways, given that their elevations, geologies, topographies, vegetations and climate forcings are very different, even if they are within the same basin.

Wong, Jefferson Razavi, Saman Bonsal, Barrie Wheater, Howard Asong, Zilefac Elvis. (2017). Inter-comparison of daily precipitation products for large-scale hydro-climatic applications over Canada. Hydrology and Earth System Sciences. 21. 2163-2185. 10.5194/hess-21-2163-2017.

Furthermore, many ecozones are split among several basins. The Montane Cordillera stations are divided among the Nelson, Peace-Athabasca, and Fraser basins. These stations would be expected to show some similarities, although local conditions would also apply.

It would be very useful to have the ecozones superimposed on the maps. It would also be useful to take the ecozones into account when grouping the analyses.

Line 338

"...the Arctic has the least diversity in the streamflow regime. All considered 12 streams are associated with large degrees to glacial regime, out of which five and six streams show increasing and decreasing trends in the membership, respectively."

The fact that half of the streams in the basin change in each direction is confusing. Does this imply that the changes are not a result of climate shifts, but rather of short-duration weather trends? Or is it that the streams are in different climatic zones?

It would be very useful to have a map, or maybe more than one map, of the sites showing their changes in regime type. This would allow the reader to see if the changes are spatially related. Again, it would be very useful to have the ecozones superimposed.

The "glacial" type is problematic. Looking at Figure 6, at least 16 of the "glacial" basins cannot include any glaciers at all, as they are not in mountains. No doubt many of the mountain basins do not contain glaciers, either. The same issue is true of the "niveo-glacial" type. I understand that the authors are using the term "glacial" to refer to the shape of the cluster's annual hydrograph, but the term is confusing. Worse, the authors are grouping together streams with very different causes for their behaviours.

The source of the archetypal "glacial" stream, Kazan River above Kazan Falls, is in northern Saskatchewan, where there are no glaciers. Looking at Figure 7, the main difference between the "glacial" and "nivo-glacial" types would appear to be that the former has a shallower recession limb. According to en.wikipedia.org/wiki/Kazan_River:

"The river headwaters are in northern Saskatchewan[7] at Kasba Lake... Along its course the river flows through several lakes, including Ennadai Lake and Yathkyed Lake."

So the cause of the shallow recession limb is almost certainly storage within the lakes in the basin.

Line 245:

"Architype (sic) streams are those streams that have the highest association to the

identified regime types and can represent the characteristics of a given regime better than other members of the cluster."

As the Kazan River is controlled by lakes, it would be very difficult to transition to another cluster type. I see that many of the "glacial" and "niveo-glacial" streams lie within the Canadian Shield. Are many of these also dominated by lakes?

Where an unglaciated stream transitions between "glacial" and "niveo-glacial" types, as in the Hudson Bay and Arctic Seaboard basins, it cannot represent a change in the glacial contribution. It is therefore important to separate those basins containing glaciers, from those which do not. Where there are glaciers, and the stream transitions from "niveo-glacial" to "glacial", would this imply an increase in glacial contributions? If so, would this be justified by what we understand about glacial hydrology?

Line 244

"Figure 5 summarizes the results, showing c = 6 as the optimal number of clusters."

I think this needs to be explained in more detail. Why is 6 the optimum number of clusters? I can see that the indices become quite flat around c = 6, but what do the indices mean, i.e. are small index values better (this is not explained)? If so, why not use c = 7, as it looks to be slightly better for the Separation Index and the Xie and Beni Index? Is there a reason why it is advisable to use fewer clusters?

Figure 7 is useful to demonstrate the differences among the clusters. It would be extremely useful to see similar plots indicating the cluster transitions. For example, what does it look like when the streams transition from "glacial" to "niveo-glacial", or vice-versa? Because so many climate signals are used, it is not easy to see how the changes in the hydrograph relate to the transition from one cluster to another.

Technical comments

How were the calculations performed? I assume that some software was used. It should be credited and described. If possible, the software should be made available

for others to test and use.

I believe that "architype" is a misspelling of "archetype"

Figure 5

The x axis labels are misspelled – "Numer" should be "Number".

Figure 7

In the interests of space, it would be a good idea to omit the periods in the x-axis label month names, and also the x-axis title "Month". The caption refers to the "expected" annual hydrograph. What does this mean? Are these the mean (or median) weekly values? The y-axis label is in "mm/week$^{-1}$", i.e. in mm x week. Obviously this is incorrect.

---

## Referee Comment (RC2) · Anonymous Referee #2 · 8 Oct 2020

The paper presented by Zaerpour and colleagues proposes a new framework for identifying shifts in streamflow regimes with a subsequent application to Canadian streamflow. While the methodical challenge and the chosen case study are clearly intersting, the convoluted nature of both the methodology and the presentation of the result prevent me from recommending the paper for publication in its present form. Below, I summarize my main points that need to be clarified.

A: METHODOLOGY.

A.1 Comment on clustering and change detection: The authors propose to use a fuzzy clustering algorithm to first identify groups of stations (or degree of membership of each

station to each group) within a matrix of Indicators of Hydrologic Alterations (IHA) for a given – short – time window. Subsequently the degree of membership if each station to each class is computed for further time-frames. Overall, I can follow this approach and on first reading the description of the methodology makes sense (note a minor issue mentioned below). However, I have several questions regarding the choice of this particular approach and the stability of the analysis:

Question A.1.1: How stable is the estimate? The results of the analysis are crucially dependent on the identification of clusters in the first period. However, the streamflow climatologies (or IHAs) used for estimating these clusters are only computed using a small fraction of the available data, which is likely to yield unstable estimates. As a consequence investigating shifts in these clusters may be confounded by estimation errors. For example, I wonder if the authors would reach the same conclusions if clusters would have been identified using another period. Since the paper does not report on the stability of the estimate, it is hard to evaluate whether the overall conclusions are affected by the arbitrary choice of the first time window for identifying clusters (e.g. why not use the last window or one in the centre). Approaches for combatting this issue could be to (a) use all time windows for identifying the clusters and subsequently assessing how the degree of membership of each station to each cluster changes over time or (b) repeating the analysis with clusters identified for each time window and report the associated spread.

Question A.1.2: What is the benefit over a classical EOF/PCA analysis? Technically the analysis has distinct similarities to applications of dimension reduction methods such as Principal Component Analysis (PCA)/Empirical Orthogonal Function (EOF) analysis or Multidimensional Scaling (MDS) to spatially distributed time series. Of course, these methods do not evolve around the idea of "clusters" but identify modes of similar variability, but the strategy to first identify a membership matrix (analogue to "leading EOF patterns") which are then projected onto individual stations. In the EOF/PCA world, an analogue approach would yield a filtered time series at each station in which then

again could be used to assess regime shifts without the need of developing a new (and somewhat convoluted methodology).

A.2 Why use a combination of Kendall's tau and R2 for the attribution work? To me it appears to be a bit convoluted to use two very different metrics (Kendall's tau and R2) for the attribution work. While Kendall's tau operates on ranks and is thus less sensitive to non-linarites or outliers, R2 is in essence a linear metric. As an alternative single metric I could e.g. imagine to rely on Spearmans rank-correlation coefficient together with a simple test of significance thereof.

A.3 Why not just look at trends in time series of monthly means and timing indicators? The analysis revolves around what the authors refer to as Indicators of Hydrologic Alterations (IHA). These are essentially a: The annual mean. b: the mean of each month and c: the timing of low/high flows. While reading I wondered if it would not have been sufficient to simply show maps of the trends of each of these metrics to arrive at the same conclusions?

A.4 Why stratify the analysis along large drainage basins. While I acknowledge the tradition of stratifying the analysis of streamflow data along drainage basins I wonder if this is the ideal choice in this particular instance. The regime classes identified by the authors essentially reflect different climatological regions (e.g. colder, snow dominated vs. warmer, rainfall dominated). Continental-scale drainage basins typically cover large climatic gradients and apart from the case where stations are hydrologically connected (how many of them are?), we would not expect a-priory the drainage basin would have much explanatory power on the climatology. Alternatively, I could imagine an assessment of changes in the underlying climate drivers (e.g. temperature, precipitation) would contribute to a deeper understanding of the associated changes.

B: PRESENTATION

Overall, I found the paper to long, a bit convoluted and therefore cumbersome to read. Some reasons:

1. The key selling point advertised in the title (i.e. the methodology) is featured in about 10% (4 of 39 pages) of the article and the properties of the methodology (e.g. stability or relation to alternative techniques) are neither assessed and nor discussed.

2. The description of the results is very convoluted and I found it difficult to extract the key message upon first reading. For example, I would value if the results description would focus on overarching patterns/conclusions instead of a diligent, but lengthy description of details.

3. I found figures 8,9,10,11 quite hard to assess on first reading. Would it be possible to summarize these results e.g. in sets of 6 maps (one for each cluster).

C: MINOR ISSUES

1. There is a significant number of grammatical mistakes (i.e. missing articles) in the paper that need to be resolved by the authors.

2. Text following equation 2c: V (matrix of centroids) is described twice. This indicates that the methods section might have been written in a sloppy manner, raising the question if everything is correct. I did not have the time to check all the indices etc. in detail.

3. How is the "timing of annual low/high flow" defined? Is it the day of year of the smallest/largest value? If yes: how is the discontinuity between day 365 and day 1 handled?

---

## Author Comment (AC1) · 24 Nov 2020

**A novel algorithmic framework for identifying changing streamflow regimes: Application to Canadian natural streams (1966-2010)**

**Point-to-point Reply to Anonymous Reviewer 2**

Masoud Zaerpour, Shadi Hatami, Javad Sadri, and Ali Nazemi

We greatly acknowledge the time and effort dedicated to evaluation of our manuscript by the Anonymous Reviewer 2 (AR2). We have reflected deeply on AR2's thoughtful comments, and are ready now to provide our point-to-point responses as of the following. In this response letter, comments provided by the AR2 are numbered and listed in the same order received and are shown in *Italic*. Our responses are given in Plain texts. Please note that some of the comments received discuss multiple issues. We attempt to address each of these issues individually by separating them from one another.

1. The paper presented by Zaerpour and colleagues proposes a new framework for identifying shifts in streamflow regimes with a subsequent application to Canadian streamflow. While the methodical challenge and the chosen case study are clearly interesting, the convoluted nature of both the methodology and the presentation of the result prevent me from recommending the paper for publication in its present form. Below, I summarize my main points that need to be clarified.

**Response:** Many thanks for your thoughtful and positive review. We are grateful for receiving your comments and have done all of our responsible efforts to address them in the best way possible. In the revised manuscript, we explained our methodology in more details and clarified points raised by you and Anonymous Reviewer 1 (AR1). Since we shifted the discussion now from drainage basins into ecozones, according to a series of comments we received from you as well as AR1, the presentation of our results and framing our discussions have been majorly changed. Furthermore, we added a section to address your legitimate concern regarding the stability of the clustering results to chosen timeframes. We believe our revised manuscript, carefully evolved based on your comments and AR1's, is much improved.

**Part I. Methodology**

2. Comment on clustering and change detection: The authors propose to use a fuzzy clustering algorithm to first identify groups of stations (or degree of membership of each station to each group) within a matrix of Indicators of Hydrologic Alterations (IHA) for a given – short – time window. Subsequently the degree of membership if each station to each class is computed for further timeframes. Overall, I can follow this approach and on first reading the description of the methodology makes sense (note a minor issue mentioned below). However, I have several questions regarding the choice of this particular approach and the stability of the analysis:

**Response:** We appreciate AR2's attention to the methodological aspect of our study. We have added more detailed explanations in the methodology section to clarify the detailed questions you pointed in the following. We have also majorly extended on Section 5.2 and included a brand new analysis for the stability of the results to the choice of the timeframe. Our key findings in this regard are shared below.

3. How stable is the estimate? The results of the analysis are crucially dependent on the identification of clusters in the first period. However, the streamflow climatologies (or IHAs) used for estimating these clusters are only computed using a small fraction of the available data, which is likely to yield unstable estimates. As a consequence investigating shifts in these clusters may be confounded by estimation errors. For example, I wonder if the authors would reach the same conclusions if clusters would have been identified using another period. Since the paper does not report on the stability of the estimate, it is hard to evaluate whether the overall conclusions are affected by the arbitrary choice of the first time window for identifying clusters (e.g. why not use the last window or one in the centre). Approaches for combatting this issue could be to (a) use all time windows for identifying the clusters and subsequently assessing how the degree of membership of each station to each cluster changes over time or (b) repeating the analysis with clusters identified for each time window and report the associated spread.

**Response:** Many thanks for your thoughtful comment on the issue of stability. We did look into the uncertainty in our results, although from a different angel, and investigated the stability of the result to the length of timeframe in Sect. 5.2, particularly through **Figure S1** in the supplement. Figure S1 clearly shows that the centers of the clusters do not change significantly by the length of timeframe. While the uncertainty of our results due to the length of the timeframe was investigated thoroughly (please see the rest of discussion in Sect. 5.2 as well as the materials provided in the supplement), your comment have opened up a new way of looking at the stability of our results. By performing your suggested sensitivity analysis, see **Figure R1** below, it is now clear that cluster centers do not significantly change by altering the decadal timeframe in which the clustering is made. In Figure R1, we used all possible decadal timeframes throughout the study period to recalculate the cluster centers. The dots show the centers of clusters scaled into two dimensions using Multidimensional Scaling (MDS; Cox and Cox, 2008). Black crosses show the centers of the first decadal timeframe. The result shows that distinctions between clusters are clearly maintained, despite changing the decadal timeframe chosen for identifying the cluster centers.

Nonetheless, we feel necessary to mention that the choice of the first decadal timeframe is not arbitrary in our work, as we do a formal trend analysis on the membership values. Obviously in this context and for the purpose of understanding the evolution in the streamflow regime using the trend analysis, we should start from the first timeframe and finish in the last one and go systematically throughout all other possible timeframes in between. Our moving window methodology is particularly designed to address this.

Cox M. and Cox T.: Multidimensional Scaling. In: Handbook of Data Visualization. Springer Handbooks Comp.Statistics. Springer, Berlin, Heidelberg. https://doi.org/10.1007/978-3-540-33037-0\_14, 2008.

**Figure R1.** The stability of the cluster centers to the choice of decadal timeframe for clustering. The dots show the two dimensional scaling of the cluster centers based on the relative distance of cluster centers from one another. Black crosses show the centers identified by choosing the first decadal timeframe.

4. What is the benefit over a classical EOF/PCA analysis? Technically the analysis has distinct similarities to applications of dimension reduction methods such as Principal Component Analysis (PCA)/Empirical Orthogonal Function (EOF) analysis or Multidimensional Scaling (MDS) to spatially distributed time series. Of course, these methods do not evolve around the idea of "clusters" but identify modes of similar variability, but the strategy to first identify a membership matrix (analogue to "leading EOF patterns") which are then projected onto individual stations. In the EOF/PCA world, an analogue approach would yield a filtered time series at each station in which then again could be used to assess regime shifts without the need of developing a new (and somewhat convoluted methodology).

**Response:** Many thanks for your comment. As you noted, the key conceptual difference in our methodology is the consideration of intersected clusters to describe regime types. Through the use of our clustering-based approach, regime types can be identified using empirical data in a fully self-organizing manner and we are able to measure how degrees of belongingness to each cluster change through trend analysis. The beauty of our methodology is in its absolute transparency and the fact that transitions in both regime types (clusters) and their corresponding hydrograph shapes and/or IHAs are fully traceable and can be explicitly linked to one another. Although approaches such as PCA/EOF are informative, they cannot provide such an opportunity. For example, components in PCA analysis are combinations of several IHAs, whereas we directly link the changes in clusters to individual IHAs and measure the strength of such links using rates of shift and the coefficient of determination. Unlike the EOF/PCA approach, which assesses the shifts in each individual stations, combining the clustering algorithm with the moving-window technique provides an opportunity to identify the shifts in the flow regime across the whole country in a fully integrated way using a set of specific reference points, i.e., cluster centers in the first time episode. We believe these strengths and added benefits justify the use of our proposed methodology although it might not seem as straightforward as the use of a method such as PCA. We discussed these points in the revised manuscript.

5. Why use a combination of Kendall's tau and R2 for the attribution work? To me it appears to be a bit convoluted to use two very different metrics (Kendall's tau and R2) for the attribution work. While Kendall's tau operates on ranks and is thus less sensitive to non-linarites or outliers, R2 is in essence a linear metric. As an alternative single metric I could e.g. imagine to rely on Spearmans rank-correlation coefficient together with a simple test of significance thereof.

**Response:** Many thanks for your comment. We did not combine these two metrics nor the concept behind the Kendall's tau dependence and coefficient of determination. In fact, we use these two metrics for two different purposes: On the one hand by using Kendall's tau, we identify the sign and significance of dependencies between changes in membership degrees and changes in streamflow characteristics. On the other hand, we use coefficient of determination,  $R^2$ , to quantify how much of the variability in a given set of membership degrees can be described by changes in a specific streamflow characteristics. By using these two measures together, we not only provide a formal approach to assess the dependencies between changes in membership degrees and streamflow characteristics (through the use of Kendall's tau with formal *p*-value), but also we can facilitate quantitative communication of the impact of changes in a specific streamflow characteristic membership to another. We added this clarification in our revised manuscript.

6. Why not just look at trends in time series of monthly means and timing indicators? The analysis revolves around what the authors refer to as Indicators of Hydrologic Alterations (IHA). These are essentially a: The annual mean. b: the mean of each month and c: the timing of low/high flows. While reading I wondered if it would not have been sufficient to simply show maps of the trends of each of these metrics to arrive at the same conclusions?

**Response:** Many thanks for your comment. There are two issues here. First, while we indeed looked into the expected values of annual and monthly mean flows as well as timings of low/high flows, we also looked into the variability of IHAs as well. While the majority of current literature limit the analysis of change to expected values of IHAs, there are strong evidences, particularly in Canada that changes in the variability of streamflow characteristics can be as important as changes in the expected values – please see Table 4 in our submitted manuscript for some of the already established findings.

Second, while looking at the individual trends in mean and variability of IHAs can be informative, it has certain limitations. First and foremost, looking solely at the trends does not provide any information on how the combination of streamflow characteristics can lead into formation of specific regime type. In addition, while it would be very difficult to look into trends in 30 different characteristics, we only look into the trends in six membership values and we know that an increasing trend in membership of one regime type will inevitably translate to decreasing trends in membership values for at least another cluster. Finally, through the use of our proposed approach, it would be possible to formally relate the changes in the streamflow characteristics to changes in regime types. None of these would have been possible by looking at the simultaneous trends in the individual streamflow characteristics only.

7. Why stratify the analysis along large drainage basins. While I acknowledge the tradition of stratifying the analysis of streamflow data along drainage basins I wonder if this is the ideal choice in this particular instance. The regime classes identified by the authors essentially reflect different climatological regions (e.g. colder, snow dominated vs. warmer, rainfall dominated). Continental-scale drainage basins typically cover large climatic gradients and apart from the case where stations are hydrologically connected (how many of them are?), we would not expect a-priory the drainage basin would have much explanatory power on the climatology. Alternatively, I could imagine an assessment of changes in the underlying climate drivers (e.g. temperature, precipitation) would contribute to a deeper understanding of the associated changes.

**Response:** Many thanks for your insightful comment. A similar comment was also given by AR1 regarding the suitability of the basin/sub-basin system in discussing/framing our results. Accordingly, we majorly revised our manuscript and considered ecozones (Wiken, 1986; Lespinas et al., 2015) as units in which we frame our results, as oppose to the drainage basins/sub-basins we used in our initial submission. This effort has significantly improved the presentation and interpretation of our results. We have also considered climate regions provided by Environment and Climate Change Canada, but we figured that ecozones provide the most suitable unit for discussing our results, as ecozone not only considers climatic factors but also geology, soil characteristics, vegetation, topography, etc. (Wong et al., 2017). Just as an example, Figures R2 and R3 as well as Table R1 below show the distributions of the considered RHBN streams as well as the result of our clustering analysis presented at ecozone scales.

Regarding the hydrological connectivity of the considered RHBN streams, we did a rigorous analysis over

all selected stations to figure out if the stations are connected. For this purpose, we used the HYDAT data along with the National Hydrographic Network (NHN) provided by Natural Resources Canada to determine the streamflow network and the flow direction at each sub-basin. Accordingly, we realized that there is only one pair of stations (i.e., 01AD002 and 01AD003) located in Saint John- St. Croix sub-basin that are hydrologically connected. To keep the consistency of our analysis, we will exclude 01AD003 from our revised manuscript, so that we can come up with 105 streams that are hydrologically independent from one another.

Lespinas, F., Fortin, V., Roy, G., Rasmussen, P., & Stadnyk, T.: Performance evaluation of the Canadian precipitation analysis (CaPA). Journal of Hydrometeorology, 16(5), 2045-2064, https://doi.org/10.1175/JHM-D-14-0191.1, 2015.

Wiken, E.B.: Terrestrial Ecozones of Canada. Ecological Land Classification, Series No. 19. Environment Canada. Hull, Quebec. pp. 26, 1986.

Wong, J. S., Razavi, S., Bonsal, B. R., Wheater, H. S., and Asong, Z. E.: Inter-comparison of daily precipitation products for large-scale hydro-climatic applications over Canada, Hydrol. Earth Syst. Sci., 21, 2163–2185, https://doi.org/10.5194/hess-21-2163-2017, 2017.

**Table R1.** List of Canadian ecozones with at least one RHBN station in this study, along with their abbreviations and the number of RHBN stations considered within each ecozone.

| Abbreviation | Ecozones          | # of stations | Abbreviation | Ecozones             | # of stations |
|--------------|-------------------|---------------|--------------|----------------------|---------------|
| EZ2          | Northern Arctic   | 1             | EZ8          | Mixedwood Plains     | 5             |
| EZ3          | Southern Arctic   | 1             | EZ9          | Boreal Plains | 6             |
| EZ4          | Taiga Plains      | 1             | EZ10         | Prairies             | 2             |
| EZ5          | Taiga Shield      | 4             | EZ12         | Boreal Cordillera    | 7             |
| EZ6          | Boreal Shield     | 25            | EZ13         | Pacific Maritime     | 9             |
| EZ7          | Atlantic Maritime | 26            | EZ14         | Montane Cordillera   | 19            |

Figure R2. The distribution of the selected 106 RHBN streamflow stations within the Canadian ecozones.

---

## Author Comment (AC2) · 24 Nov 2020

**A novel algorithmic framework for identifying changing streamflow regimes: Application to Canadian natural streams (1966-2010)**

**Point-to-point Reply to Anonymous Reviewer 1**

Masoud Zaerpour, Shadi Hatami, Javad Sadri, and Ali Nazemi

We greatly acknowledge the time and effort dedicated to evaluation of our manuscript by the Anonymous Reviewer 1 (AR1). We have reflected deeply on the thoughtful comments given by AR1, and now are ready to provide our point-to-point responses as of the following. In this response letter, comments provided by the AR1 are numbered and listed in the same order received and are in *Italic* fonts. Our responses are given in Plain text. Please note that some of the comments received discuss multiple issues. We attempt to address each issue individually by separating them from one another.

**Part I. General comments**

1. The analyses presented are interesting, and may be useful in establishing the changes in the regimes of Canadian rivers. Unfortunately, the results are undermined by their presentation. With some revisions, I believe that the paper can make a good contribution.

**Response:** We highly appreciate the positive and extremely constructive review comments given by the AR1. We are grateful that AR1 has found our work interesting and with practical merits.

2. The writing needs quite a bit of editing. There are too many grammatical mistakes to list here, and the writing is often unclear.

**Response:** Many thanks for your comment. We do acknowledge your concern. We have rigorously edited our paper to avoid grammatical mistakes. The revised version of our manuscript is now much improved. We also plan to ask a professional English editor to review our revised manuscript prior to submission to the journal.

3. There are missing articles in many of the sentences, such as the first one: Page 1, Line 28 "Natural streamflow characteristics have been critical consideration". This is sentence is missing the article "a" before "critical", or needs to make "consideration" a plural

**Response:** Many thanks for your comment. This is taken care of in the revised paper.

4. In many sentences there are disagreements in number, i.e. between singular and plurals: Page 2, Line 34 "some others determines"

**Response:** Many thanks for your comment. This is taken care of in the revised paper.

5. As was stated above, the writing is often unclear, as in the caption of Figure 9: "Figure 9. Mapping shifts in natural streamflow throughout Canada during 1966 to 2010. Rates of shift among various regime types in each stream are shown by shades of grey that quantifies how much decline in the giver regimes shown in the x-axes in each panel can result into incline in the receiver regime type corresponding with the column in which the panel is located. Columns filled with diagonal lines show the identical regime types with the receiving regimes identified in the column where the panel is located."

**Response:** Many thanks for your comment. This is now taken care of. Please note that in the quest for more clarity, we have revised Figure 9 with a set of six maps. Please see the **Figure R5 below** in response to the comment #9.

**Part II. Specific comments**

6. There appears to be only one gauging station within the Canadian Prairies. This is disappointing as the hydrology of the region is very important and has seen many effects of changes in climate. There are several RHBN stations within the prairies, according to this website https://www.canada.ca/en/environment-climatechange/services/wateroverview/quantity/monitoring/survey/data-products-services/reference hydrometric-basin-network.html.

Whitfield et al. (2020) grouped responses of streams into 3 clusters in the Prairies and adjacent areas, using a very different clustering methodology. I assume that there were no other prairie stations which met the authors' criteria. However, it would be good to have this explained. Would the use of a slightly different analysis period have allowed the inclusion of more prairie streams?

Whitfield, P.H., Shook, K.R., Pomeroy, J.W., 2020. Spatial patterns of temporal changes in Canadian Prairie streamflow using an alternative trend assessment approach. Journal of Hydrology 582, 124541. https://doi.org/10.1016/j.jhydrol.2020.124541

**Response:** Many thanks for your comment. In the first step of our study, we did a rigorous analysis to accommodate as many RHBN stations as possible with the longest common period. In fact, as noticed also by Whitfield et al. (2020), the Prairie region does not include many RHBN stations with long-term and continuous data record. Unfortunately, we could find only two stations in the prairie region during 1966 to 2010 that met our data criteria, i.e., having a continuous daily record with less than ONE month worth of missing data in a typical hydrologic year. To address your comment, we altered our data period to 1976-2010 and repeated our search for new stations in the Prairie region and accordingly our clustering analysis. This new effort has resulted into consideration of NINE new stations in the Prairie region. We compared the result of our new analysis with the results we obtained with previously selected RHBN stations in the Prairie region, namely, Waterton River near Waterton Park (05AD003) and Belly River near Mountain View (05AD005) during 1976-2010. Figure R1 below summarizes our findings. Left panel shows the clustering results related to the new NINE stations, i.e. S1 to S9 in comparison with the two previously selected stations, i.e. 05AD003 and 05AD005. The right panel shows the analysis of trends in anomalies of decadal memberships, in which the stations in each ecozone are sorted from the lowest to highest elevations from the top to the bottom. This analysis shows that the three stations in the southwest of prairie are mostly of type C2 (i.e., fastresponding/summer peak regime; previously named as Naivo-glacial – see our response to your 10th comment below). The station S8 in the northwest and S1 in the east of prairie are mostly belong to C1 (i.e., slowresponding summer regime – previously named as Glacial). The rest of stations are mostly belong to the C3 and C4. The analysis of trends in anomalies of memberships shows mainly decreasing trends in belongingness to C1 and C2 clusters and increasing trend for C5 and C6. This analysis will be included in our revision as a part of our Discussion section.

**Figure R1.** Validation of the proposed clustering approach in the Canadian Prairies using nine new and two previouslyselected RHBN stations during 1976 to 2010. The color bar in the left map show the degrees of membership to each cluster. The right panel show the trends in the degree of membership in the six clusters at the considered 11 stations. Positive and negative trends are shown with red and blue colors, respectively. Sharp colors show significant cases. New stations S1 to S9 are sorted from the lowest to the highest elevations from the top to the bottom.

7. Although hydrologists are used to working with river basins, grouping the stations by basin is not always useful. As shown in Table 2, Canadian river basins are very large. Wong et al. (2017) identify 15 ecozones in Canada, many of which are spanned by single basins. For example, the Nelson River system spans the Montane Cordillera, Prairies, Boreal Plains, Canadian Shield and the Hudson Plain. Stations in differing ecozones would not be expected to behave in similar ways, given that their elevations, geologies, topographies, vegetations and climate forcings are very different, even if they are within the same basin.

Wong, Jefferson Razavi, Saman Bonsal, Barrie Wheater, Howard Asong, Zilefac Elvis. (2017). Intercomparison of daily precipitation products for large-scale hydro- climatic applications over Canada. Hydrology and Earth System Sciences. 21. 2163- 2185. 10.5194/hess-21-2163-2017.

Furthermore, many ecozones are split among several basins. The Montane Cordillera stations are divided among the Nelson, Peace-Athabasca, and Fraser basins. These stations would be expected to show some similarities, although local conditions would also apply. It would be very useful to have the ecozones superimposed on the maps. It would also be useful to take the ecozones into account when grouping the analyses.

**Response:** Many thanks for this very insightful comment. We have done a major revision in our manuscript to address your comment and a similar comment raised by Anonymous Reviewer 2 (AR2) regarding the appropriateness of the basin/sub-basin system for integration and discussion of our results. In the revised manuscript that will be provided upon the approval of the editor, we will present and discuss our results entirely at the ecozone scale except only in positioning against earlier studies (currently Section 5.1 in the submitted manuscript). We should also mention that our revised manuscript has significantly improved and the result can be much better interpreted by moving to the ecozone scale. To just demonstrate here how

mappings look like with consideration of ecozones, below in **Figure R2** and **Table R1** (these are counterparts of Figure 4 and Table 1 in the submitted manuscript, respectively), we show how the considered RHBN stations are situated and distributed within the 15 Canadian ecozones. We have also shown in **Figure R3** the results of our clustering analysis at the ecozone scale. This is equivalent to the results we provided in Figure 6 in the submitted manuscript but this time at the ecozone scale. As it can be seen, the results are much more interpretable at ecozone scale. We highly appreciate AR1's constructive comment.

---

## Author Response (AR1)

**A novel algorithmic framework for identifying changing streamflow regimes: Application to Canadian natural streams (1966-2010)**

**Point-to-point Reply to the Editor and Reviewers' comments**

Masoud Zaerpour, Shadi Hatami, Javad Sadri, and Ali Nazemi

**I. A bird view over the revised manuscript**

We are extremely grateful for the comments received from the Editor and the two anonymous reviewers. We have done all our responsible efforts to digest, reflect, and apply the editor and reviewers' comments. Some of the comments needed substantial efforts, including a new set of simulations and analyses. Before moving to our point-to-point responses to the comments, our changes are briefly summarized in Table R1 below. We believe we have a better paper now.

**Table R1.** A brief overview on the revisions made in response to the review comments

| No. | Section | Summary of changes made |
|---|---|---|
| 1 | Title | Title is revised |
| 2 | Abstract | Abstract is shortened and a glimpse of the results at the ecozone scale is provided. |
| 3 | Sect. 1 Introduction | Introduction is substantially shortened and is much more focused. |
| 4 | Sect. 2 Methodology | New discussions are added in response to Reviewer 2 to highlight the key advantages of the proposed framework over commonly used methodologies such as EOF/PCA. In response to both reviewers, we extend the explanation of our methodology in particular the three validity indices. We add the description of the knee (elbow) method for finding the optimal number of clusters. We also include a new figure (Figure 4) to better show how our proposed algorithm works. |
| 5 | Sect. 3 Case study and data | In response to Reviewers 1 and 2, we introduce terrestrial ecozones of Canada, which is later used for the interpretation and synthesis of our key results in Sect. 4. |
| 6 | Sect. 4.1 | The section is shortened by moving a couple of figures to the supplement. New names are introduced for the six identified clusters; and the distribution of the regime types across eczones are explained. |
| 7 | Sect. 4.2 | The section is substantially revised. In response to Reviewers 1 and 2, the figure related to trend in degrees of membership is changed to a set of six maps, which helps readers to have a synoptic view of changes in the streamflows across different ecozones/clusters. New Sankey plots are included to better show the regime shift across streams and ecozones. |
| 8 | Sect. 4.3 | This section is shortened substantially in response to Reviewers 1 and 2. The two figures in the earlier version are merged in one figure and results are now discussed across ecozones. |
| 9 | Sect. 5 Discussion | In response to Reviewer 2, the analysis of uncertainty of the result to the location/length of baseline timeframe is added to Sect. 5.1. In response to Reviewer 1, NINE unseen gauging stations in the Canadian Prairies are selected as a basis to verify our earlier results in this rather overlooked ecozone. This new analysis and the related discussion are added to Sect. 5.2. |
| 10 | Sect. 6 Conclusions and outlook | The conclusion is modified based on the analysis of the results at the ecozone scale. |
| 11 | References | Despite adding few more references, the number of references is reduced substantially. |
| 12 | Supplement | Our results at basin/sub-basin scale are moved to the supplement. |

The summarize the is paper majorly revised. Below, we respond to the review comments point-to-point, and in the order received. Please note that we have numbered the comments received from each reviewer separately. Some of the comments received discuss multiple issues. We attempt to address each issue individually by separating them from one another. Reviewers comments are in **BLACK**. Our responses are in **RED**.

**II. Reply to comments received from the editor**

1. *Two reviewers have made very insightful comments to your manuscript. Thanks for the detailed responses to the points they raised. As your replies indicate the willingness for substantial revision, I would like to invite you to resubmit a revised version after major revisions.*

**Response:** Many thanks for handling our manuscript. As you noted, we received extremely constructive comments that helped us to significantly improve our manuscript as a whole.

2. *I also very much welcome the effort to include the additional prairie stations, use ecozones for reference and to change the terminology for the glacial regime seasonality and on the stability test and the suggested clarifications.*

**Response:** We are pleased that our efforts during the public response period are recognized by the editor. We added the analysis related to the prairie stations in **Sect. 5.2** of the revised manuscript. Throughout the revised manuscript, we used the ecozones instead of basins as the primary spatial scale to interpret and synthesis our results. Accordingly, the discussion of our results at the basin/sub-basin scales is now moved to the supplement. The terminology used for the cluster is now based on two key characteristics, i.e. the timing of the annual peak (i.e. cold-season, freshet, and warm-season), and the form of hydrologic response (i.e. fast- *vs.* slow-response). Please see **lines 324 to 374** in the revised manuscript. The analysis of stability in our results along with other issues related to the uncertainty in our analysis is discussed in **Sect. 5.1** in the revised manuscript.

3. *I think R2's point 4 and your response are of interest to all readers and suggest that the aims and advantages need to be communicated more effectivey. However, please don't do so by adding length to manuscript.*

**Response:** We added the advantages of the proposed method compared to other available methodologies in **Sect. 2.1** in the revised manuscript. Please see **lines 75-99** in the revised manuscript. We also pointed to the transparency of our proposed algorithm in **Sect. 2.4, lines 219-225**. We are also pleased to let you know that our revised manuscript has less words than the initial submission.

4. *Regarding R1's point 17: Data and code sharing should ideally be done outside the HESS system in repositories. While it is nice to share code, I think a good methodology description and a comment on the Matlab implementation will also be sufficient.*

**Response:** Many thanks for your clarification on this. We majorly deepened the description of the methodology and the step-by-step procedure. Please see **Sect. 2**. We did mention that all calculations are performed in MATLAB. Please see **lines 98-99** in the revised manuscript.

5. *Overall you saw from the reviewers judgements that careful revision of the text is necessary for readers to understand and appreciate the work. Please pay attention to conciseness when revising as this often helps and note that the manuscript was already found too long and hence the revisions should not make*

*it even longer and more complicated.*

**Response:** We carefully revised the manuscript by adding new materials and paying attention to the conciseness of our submission. The text is reduced by 11%.

**III. Reply to comments received from Anonymous Reviewer 1**
**Part 1. General comments**

1. *The analyses presented are interesting, and may be useful in establishing the changes in the regimes of Canadian rivers. Unfortunately, the results are undermined by their presentation. With some revisions, I believe that the paper can make a good contribution.*

**Response:** We greatly acknowledge the time and effort spent by Anonymous Reviewer 1 (AR1) in the evaluation of our manuscript. We are thankful for positive and extremely constructive comments provided by AR1 and are happy that our contribution was found interesting and practically relevant in the Canadian context.

2. *The writing needs quite a bit of editing. There are too many grammatical mistakes to list here, and the writing is often unclear.*

**Response:** Many thanks for your comment. We do acknowledge your concern. We have rigorously edited our paper to avoid grammatical mistakes. The revised version of our manuscript is now much improved.

3. *There are missing articles in many of the sentences, such as the first one: Page 1, Line 28 "Natural streamflow characteristics have been critical consideration". This is sentence is missing the article "a" before "critical", or needs to make "consideration" a plural.*

**Response:** Many thanks for your comment. This is taken care of in the revised paper.

4. *In many sentences there are disagreements in number, i.e. between singular and plurals: Page 2, Line 34 "some others determines"*

**Response:** Many thanks for your comment. This is taken care of in the revised paper.

5. *As was stated above, the writing is often unclear, as in the caption of Figure 9: "Figure 9. Mapping shifts in natural streamflow throughout Canada during 1966 to 2010. Rates of shift among various regime types in each stream are shown by shades of grey that quantifies how much decline in the giver regimes shown in the x-axes in each panel can result into incline in the receiver regime type corresponding with the column in which the panel is located. Columns filled with diagonal lines show the identical regime types with the receiving regimes identified in the column where the panel is located."*

**Response:** Many thanks for your comment. This is now taken care of. Please note that in the quest for more clarity, we have revised Figure 9 with a Sankey plot. Please see the **Figure 8** in the revised manuscript.

**Part 2. Specific comments**

6. *There appears to be only one gauging station within the Canadian Prairies.  This is disappointing as the hydrology of the region is very important and  has seen many effects of changes in climate. There are several RHBN stations within the prairies, according to this website*

*https://www.canada.ca/en/environment-climate-change/services/wateroverview/quantity/monitoring/survey/data-products-services/reference hydrometric-basin-network.html. Whitfield et al. (2020) grouped responses of streams into 3 clusters in the Prairies and adjacent areas, using a very different clustering methodology. I assume that there were no other prairie stations which met the authors' criteria. However, it would be good to have this explained. Would the use of a slightly different analysis period have allowed the inclusion of more prairie streams?*

*Whitfield, P.H., Shook, K.R., Pomeroy, J.W., 2020. Spatial patterns of temporal changes in Canadian Prairie streamflow using an alternative trend assessment approach. Journal of Hydrology 582, 124541. https://doi.org/10.1016/j.jhydrol.2020.124541*

**Response:** Many thanks for your comment. As noted in Sect. 3, when we describe the data support, we did a rigorous analysis to accommodate as many RHBN stations as possible with the longest common period. In fact, as noticed also by Whitfield et al. (2020), the Prairie region does not include many RHBN stations with long-term and continuous data records. Unfortunately, we could find only two stations in the prairie region from 1966 to 2010 that met our data criteria (i.e. having a continuous daily record with less than ONE month worth of missing data in a typical hydrologic year). These two stations are Waterton River near Waterton Park (05AD003) and Belly River near Mountain View (05AD005) Having said that, we do acknowledge your comment and we believe it is relevant and fair. Accordingly, we altered our data period in the Prairie region to 1976-2010 and repeated our search for new stations. This effort has resulted into the consideration of NINE new stations in the Prairie region. We compared the result of our new analysis with the results obtained with previously selected RHBN stations in the Prairie region during 1976-2010. This new analysis is now placed in **Sect. 5.2** in the revised manuscript. In particular, **Figure 12** summarizes our findings. The left panel shows the clustering results related to the new NINE stations, i.e. P1 to P9 in comparison with the two previously selected stations, i.e. 05AD003 (S69) and 05AD005 (S70). Please see **lines 520-544** in the revised manuscript.

7. *Although hydrologists are used to working with river basins, grouping the stations by basin is not always useful. As shown in Table 2, Canadian river basins are very large. Wong et al. (2017) identify 15 ecozones in Canada, many of which are spanned by single basins. For example, the Nelson River system spans the Montane Cordillera, Prairies, Boreal Plains, Canadian Shield and the Hudson Plain. Stations in differing ecozones would not be expected to behave in similar ways, given that their elevations, geologies, topographies, vegetations and climate forcings are very different, even if they are within the same basin. Wong, Jefferson Razavi, Saman Bonsal, Barrie Wheater, Howard Asong, Zilefac Elvis. (2017). Inter-comparison of daily precipitation products for large-scale hydro- climatic applications over Canada. Hydrology and Earth System Sciences. 21. 2163- 2185. 10.5194/hess-21-2163-2017.*
   *Furthermore, many ecozones are split among several basins. The Montane Cordillera stations are divided among the Nelson, Peace-Athabasca, and Fraser basins. These stations would be expected to show some similarities, although local conditions would also apply. It would be very useful to have the ecozones superimposed on the maps. It would also be useful to take the ecozones into account when grouping the analyses.*

**Response:** Many thanks for this very insightful comment. A similar comment was raised by Anonymous Reviewer 2 (AR2) regarding the effectiveness of basin/sub-basin system for discussing/synthesizing our results. We have majorly revised our manuscript in response to this comment. In the revised manuscript, we discuss our results primarily at the ecozone scale throughout the paper, except in positioning against earlier studies (Sect. 5.3 in the revised manuscript), and moved the discussion related to the basin/sub-basin into the supplement. To just demonstrate here how mappings of stations look like with consideration of ecozones, **Figure 5** and **Table 2** in the revised manuscript show how the considered RHBN stations are situated and

distributed within the 15 terrestrial ecozones of Canada. Also, **Figure 6** in the revised manuscript show the clustering results at the ecozone scale. The rest of our analysis is also at the ecozone scale. Applying your comment has significantly improved the discussion and the interpretation of our results.

8. *Line 338 "...the Arctic has the least diversity in the streamflow regime. All considered 12 streams are associated with large degrees to glacial regime, out of which five and six streams show increasing and decreasing trends in the membership, respectively." The fact that half of the streams in the basin change in each direction is confusing. Does this imply that the changes are not a result of climate shifts, but rather of short- duration weather trends? Or is it that the streams are in different climatic zones?*

**Response:** Many thanks for another constructive comment. After performing the analysis of trends, based on your previous comment in ecozones, it became clear that different directions of a trend in one drainage basin largely correspond to different ecozones – see **Figure 7** in the revised manuscript. Specific to your comment, Figure 7clearly shows that degrees of membership to cluster C1 (named previously glacial, in the revised version named slow-response/warm-season peak) decline in Shield, but increase in the Boreal Plains. While the ecozone classification largely determines directions of trends inside each drainage basin, further variations in directions of trends can be observed inside each ecozone. These variations correspond to elevation and latitude. This is highlighted in several spots in the paper and supported by evidence – see e.g. **Figure S5** in the revised Supplement. This figure is the same as Figure 8 in the earlier version of the manuscript (currently Figure S4 in the revised Supplement), but rows are rearranged based on ecozones, and are sorted in each ecozone by elevation.

9. *It would be very useful to have a map, or maybe more than one map, of the sites showing their changes in regime type. This would allow the reader to see if the changes are spatially related. Again, it would be very useful to have the ecozones superimposed.*

**Response:** Many thanks for your comment. As mentioned, we majorly revised the presentation and discussion of our result at the ecozone scale. To address your concern, both **Figures 6 and 7** in the revised are now presented as maps, superimposed on ecozones. Please also see **Figure S6** in the revised supplement. To better facilitate the communication of transition from one regime type to others, we also added a set of Sankey diagrams in **Figure 8**, showing how the streamflow regime in our considered RHBN streams, grouped based on ecozones, transform to a specific regime type.

10.     *"glacial" type is problematic. Looking at Figure 6, at least 16 of the "glacial" basins cannot include any glaciers at all, as they are not in mountains. No doubt many of the mountain basins do not contain glaciers, either. The same issue is true of the "niveo- glacial" type. I understand that the authors are using the term "glacial" to refer to the shape of the cluster's annual hydrograph, but the term is confusing. Worse, the authors are grouping together streams with very different causes for their behaviours.*

**Response:** This is another great comment from AR1 that we highly acknowledge. Yes! Our initial idea behind the naming of our clusters was to point to the shape of the annual hydrograph rather than the source from which streams are initiated from. To address your very relevant comment and to avoid any potential confusion, we changed the naming of our regime types to exclusively point at the shape without any referral to the source and/or the cause of the streamflow. Our new naming system is based on two key characteristics, namely the timing of the peak (i.e. cold-season peak, freshet peak, and warm-season peak) and the form of hydrologic response (i.e. fast *vs.* slow). The hydrologic response can be identified by streamflow variability, the higher the variance is for an annual streamflow hydrograph, the faster the streamflow response is. Using this new system, our previously termed "glacial" regime will be now "slow-response/warm-season peak", comprising streams with very strong seasonality and high discharge in summer and relatively less variability

in annual streamflow hydrograph compared to the C2, which was previously termed as the "nivo-galacial" and now termed as "fast-response/warm-season peak" regime. The third cluster (C3) was previously termed as the "nival" regime is now named as "slow-response/freshet peak" regime and includes streams in which the annual streamflow volume is mainly contributed by the spring snowmelt with relatively less variation in the shape of hydrograph compared to C4, the "fast-response/freshet peak" regime (previously named as the "nivo-pluvial" regime). C5 regime, previously termed as the "pluvio-nival regime", is now named as the "slow-response/cold-season peak" regime, which comprises streams with a weak seasonality and slightly higher discharge in fall and winter seasons. C6, previously termed as "pluvial" is now named the "fast-response/cold-season peak" regime, showing higher variation in shapes of decadal hydrographs. We believe applying this comment has considerably improved the clarity of our revised manuscript.

11.    *The source of the archetypal "glacial" stream, Kazan River above Kazan Falls, is in northern Saskatchewan, where there are no glaciers. Looking at Figure 7, the main difference between the "glacial" and "nivo-glacial" types would appear to be that the former has a shallower recession limb. According to en.wikipedia.org/wiki/Kazan_River: "The river headwaters are in northern Saskatchewan[7] at Kasba Lake... Along its course the river flows through several lakes, including Ennadai Lake and Yathkyed Lake." So the cause of the shallow recession limb is almost certainly storage within the lakes in the basin.*

**Response:** Many thanks for your comment. You are absolutely right! We believe that with the new naming suggested above, we avoided this problem, as C1 regime, i.e., "slow response/warm-season peak", can refer to the streams initiated from both glaciers and lakes. In addition, we separated the glacier-fed stations from lake-dominated. Please see **Figure R1** below.

[Figure]

**Figure R1.** Streams belonging to "slow-response/warm-season peak" regime (C1) are distinguished based on whether they are glacier-fed (blue circles) or and non-glaciated streams (red squares).

12.    *Line 245: "Architype (sic) streams are those streams that have the highest association to the identified regime types and can represent the characteristics of a given regime better than other members of the cluster." As the Kazan River is controlled by lakes, it would be very difficult to transition to another cluster type. I see that many of the "glacial" and "niveo-glacial" streams lie within the Canadian Shield. Are many of these also dominated by lakes?*

**Response:** Thanks again for this careful comment. In fact, 14 out of 16 stations located in the Canadian Shield are indeed lake-dominated. Please see **Figure R1** above and the discussion below. We believe that we had a terminology issue and now that we change the naming of the flow regime, this issue should be resolved. For lake-dominated streams, moving from C1 to C2 means having more variability in the annual streamflow hydrograph, revealing faster response – or more dynamics – in the streamflow.

13. *Where an unglaciated stream transitions between "glacial" and "niveo-glacial" types, as in the Hudson Bay and Arctic Seaboard basins, it cannot represent a change in the glacial contribution. It is therefore important to separate those basins containing glaciers, from those which do not.*

**Response:** As mentioned above, this confusion should be greatly resolved now by the new naming of the regime types. Transition from C1 to C2 refers to increases in the variability in annual streamflow hydrographs. To specifically address your comment, we have used GLIMS and NSIDC Glacier Database (https://doi.org/10.7265/N5V98602), to distinguish between glacial and other stream types. **Figure R1** shows this analysis for C1, the "slow-response/warm-season" regime, in which glacier-fed streams are distinguished from lake-dominated ones.

14. *Where there are glaciers, and the stream transitions from "niveo-glacial" to "glacial", would this imply an increase in glacial contributions? If so, would this be justified by what we understand about glacial hydrology?*

**Response:** Many thanks for your comment. Although we have not directly explored the role of glacial contribution by analyzing glacial coverage/storage data, we believe that transition from C2 to C1, i.e., from "fast-response/warm-season peak" to "slow-response/warm-season peak" (previously termed as "niveo-glacial" and "glacial", respectively) means more streamflow contribution from glacial retreat rather than the annual snowpack. This means that contribution from the annual snowmelt is lessened and more glacial retreats take place from glacial storage. We believe that this can make a potentially liable hypothesis, which should be investigated in more detail using glacial coverage data. This clearly remains beyond the scope of this paper and should be investigated through an independent study. Having said that, we added this hypothesis to the revised paper (see **lines 596 to 599**) and suggested further explorations toward addressing this among several other unanswered solutions.

15.    *Line 244 "Figure 5 summarizes the results, showing c = 6 as the optimal number of clusters.". I think this needs to be explained in more detail. Why is 6 the optimum number of clusters? I can see that the indices become quite flat around c = 6, but what do the indices mean, i.e. are small index values better (this is not explained)? If so, why not use c = 7, as it looks to be slightly better for the Separation Index and the Xie and Beni Index? Is there a reason why it is advisable to use fewer clusters?*

**Response:** Many thanks for this comment. We used the elbow method for determining the optimal number of clusters (please see Zhao et al., 2008; Satopaa et al., 2011; Kuentz et al., 2017). In fact, we chose c=6 as the cutoff point, because while the separation index continues to decrease slightly, the added complexity, i.e.

added number of parameters representing a new cluster, cannot be justified. Please note that adding a new cluster, in this case, means identifying 30 new parameters. When c = 6, a significant flattening takes place in the three indices, indicating an elbow. We noted that adding more clusters leads to ambiguity in the interpretation of the clusters due to less significant differences between regime types. This discussion is added to the revised manuscript. Please see **lines 174-176**.

Zhao, Q., Hautamaki, V., and Fränti, P.: Knee point detection in BIC for detecting the number of clusters. In International conference on advanced concepts for intelligent vision systems (pp. 664-673). Springer, Berlin, Heidelberg, https://doi.org/10.1007/978-3-540-88458-3_60, 2008.

Satopaa, V., Albrecht, J., Irwin, D., and Raghavan, B.: Finding a" kneedle" in a haystack: Detecting knee points in system behavior. In 2011 31st international conference on distributed computing systems workshops (pp. 166-171). IEEE, https://10.1109/ICDCSW.2011.20, 2011.

Kuentz, A., Arheimer, B., Hundecha, Y., and Wagener, T.: Understanding hydrologic variability across Europe through catchment classification, Hydrol. Earth Syst. Sci., 21, 2863–2879, https://doi.org/10.5194/hess-21-2863-2017, 2017.

16. *Figure 7 is useful to demonstrate the differences among the clusters. It would be extremely useful to see similar plots indicating the cluster transitions. For example, what does it look like when the streams transition from "glacial" to "niveo-glacial", or vice-versa? Because so many climate signals are used, it is not easy to see how the changes in the hydrograph relate to the transition from one cluster to another.*

**Response:** Many thanks for this comment. To address your comment and to better demonstrate transitions between regime types, we revised our visualization approach in different figures to explicitly illustrate how changes in regime types correspond to changes in streamflow characteristics. **Figure 4** in the revised manuscript, for example, provides a hypothetical example for transitions between clusters along with the process of attributions to changes in streamflow features. The changes in streamflow regime are shown with grey and pink envelopes in the left panels. Right panels show rates of shift, as well as the attribution of a regime shift to changes in streamflow characteristics. **Figure 8** in the revised manuscript shows how different regimes are evolving to one another in different streams and **Figure 9** in the revised manuscript shows how the dominant regime shifts are resulted from changes in streamflow characteristics.

**Part 3. Technical comments**

17. *How were the calculations performed? I assume that some software was used. It should be credited and described. If possible, the software should be made available for others to test and use.*

**Response:** Many thanks for your comment. We have developed our framework in MATLAB. In brief, the indicators of hydrological alteration are described at Ritcher et al., (1996) and coded by us. The procedure of the fuzzy clustering method is based on the algorithm proposed by Bezdek (1981) and it is carried out in MATLAB using the built-in MATLAB function *fcm*. The formulation of the three validity indices can be found in Xie and Beni (1991) and Bensaid et al. (1996) and coded by us in MATLAB. All other procedures, i.e. trend analysis, attribution, etc. are also coded in MATLAB. Based also on the Editor's comment, we indicated MATLAB as our computational platform in **lines 98-99** in the revised manuscript

Bensaid, A.M., Hall, L.O., Bezdek, J.C., Clarke, L.P., Silbiger, M.L., Arrington, J.A. and Murtagh, R.F.: Validity-guided (re) 635 clustering with applications to image segmentation. IEEE Transactions on fuzzy systems, 4(2), pp.112-123, https://doi.org/10.1109/91.493905, 1996.

Bezdek, J. C.: Pattern Recognition With Fuzzy Objective Function Algorithms, Plenum, New York, https://doi.org/10.1007/978-1-4757-0450-1_3, 1981.

Richter, B.D., Baumgartner, J.V., Powell, J. and Braun, D.P.: A method for assessing hydrologic alteration within ecosystems. Conservation biology, 10(4), pp.1163-1174, https://doi.org/10.1046/j.1523-1739.1996.10041163.x, 1996.

Xie, X.L., Beni, G.A.: Validity measure for fuzzy clustering. IEEE Trans. PAMI 3(8), 841–846, https://doi.ieeecomputersociety.org/10.1109/34.85677, 1991.

*18.    I believe that "architype" is a misspelling of "archetype"*

**Response:** Many thanks for your comment. We fixed this typo.

*19.    Figure 5: The x axis labels are misspelled – "Numer" should be "Number".*

**Response:** Many thanks for your comment. We fixed this typo.

*20.    Figure 7: In the interests of space, it would be a good idea to omit the periods in the x-axis label month names, and also the x-axis title "Month". The caption refers to the "expected" annual hydrograph. What does this mean? Are these the mean (or median) weekly values? The y-axis label is in "mm/week-1", i.e. in mm x week. Obviously this is incorrect.*

**Response:** Many thanks for your comment. In Figure 7 in the previous version (currently **Figure S3** in the supplement), the solid black and red lines are mean values of annual hydrographs during1966-1975 (grey envelopes) and 2001-2010 (pink envelopes), respectively. The term "expected" in the figure caption refers to this. As you suggested, the title of x-axis is removed and months are shown with one letter. The y-axis label is revised as "mm week$^{-1}$". Please see the revised **Figure 7**.

**IV. Reply to comments received from Anonymous Reviewer 2**

*1. The paper presented by Zaerpour and colleagues proposes a new framework for identifying shifts in streamflow regimes with a subsequent application to Canadian streamflow. While the methodical challenge and the chosen case study are clearly interesting, the convoluted nature of both the methodology and the presentation of the result prevent me from recommending the paper for publication in its present form. Below, I summarize my main points that need to be clarified.*

**Response:** We greatly acknowledge the time and effort dedicated to the evaluation of our manuscript by Anonymous Reviewer 2 (AR2). In the revised manuscript, we explained our methodology in more detail. Since we shifted the discussion now from drainage basins into ecozones, thanks to a series of extremely constructive comments received from you as well as Anonymous Reviewer 1 (AR1), we believe that the presentation of our results and framing of our discussion have been majorly improved. We also added a section to address your legitimate concerns regarding the stability of the clustering results to chosen timeframes. We believe our revised manuscript is much improved now.

**Part I. Methodology**

*2. Comment on clustering and change detection: The authors propose to use a fuzzy clustering algorithm to*

*first identify groups of stations (or degree of membership of each station to each group) within a matrix of Indicators of Hydrologic Alterations (IHA) for a given – short – time window. Subsequently the degree of membership if each station to each class is computed for further timeframes. Overall, I can follow this approach and on first reading the description of the methodology makes sense (note a minor issue mentioned below). However, I have several questions regarding the choice of this particular approach and the stability of the analysis:*

**Response:** We appreciate AR2's attention to the methodological aspect of our study. To address your comments, we have included a brand new analysis for the stability and the choice of the timeframes that we explain in more detail below.

3. *How stable is the estimate? The results of the analysis are crucially dependent on the identification of clusters in the first period. However, the streamflow climatologies (or IHAs) used for estimating these clusters are only computed using a small fraction of the available data, which is likely to yield unstable estimates. As a consequence investigating shifts in these clusters may be confounded by estimation errors. For example, I wonder if the authors would reach the same conclusions if clusters would have been identified using another period. Since the paper does not report on the stability of the estimate, it is hard to evaluate whether the overall conclusions are affected by the arbitrary choice of the first time window for identifying clusters (e.g. why not use the last window or one in the centre). Approaches for combatting this issue could be to (a) use all time windows for identifying the clusters and subsequently assessing how the degree of membership of each station to each cluster changes over time or (b) repeating the analysis with clusters identified for each time window and report the associated spread.*

**Response:** Many thanks for this thoughtful comment. Although from a relatively different angle, we did look into the uncertainty in our results and investigated how stable our findings are if the length of timeframe is changed. In **Sect. 5.1** in the revised manuscript, particularly through **Figures 9 and 10** in the revised manuscript. Figure 9 clearly shows that the centers of the clusters do not change significantly by altering the length of timeframe nor by altering the decadal timeframe in which the clustering is made. Figure 10 also shows that the result of our analysis is also quite robust and does not change by the length of the timeframes. Nonetheless, we feel necessary to mention that the choice of the first decadal timeframe is not arbitrary in our work, as we do a formal trend analysis on the membership values. Obviously in the context of trend analysis, and to understand the evolution in the streamflow regime during a certain period, we should start from the first timeframe and finish in the last one and go systematically throughout all other possible timeframes in between. Our moving window methodology is particularly designed to address this.

4. *What is the benefit over a classical EOF/PCA analysis? Technically the analysis has distinct similarities to applications of dimension reduction methods such as Principal Component Analysis (PCA)/Empirical Orthogonal Function (EOF) analysis or Multidimensional Scaling (MDS) to spatially distributed time series. Of course, these methods do not evolve around the idea of "clusters" but identify modes of similar variability, but the strategy to first identify a membership matrix (analogue to "leading EOF patterns") which are then projected onto individual stations. In the EOF/PCA world, an analogue approach would yield a filtered time series at each station in which then again could be used to assess regime shifts without the need of developing a new (and somewhat convoluted methodology).*

**Response:** Many thanks for your comment. As you noted, the key conceptual difference in our methodology is the consideration of intersecting clusters to describe regime types. Through the use of our fuzzy clustering approach, regime types can be identified using empirical data in a fully bottom-up manner; and, we are able to measure how degrees of belongingness to each cluster change through a formal trend analysis. The beauty of our methodology is in its absolute transparency: As it can be seen transitions between regime types

(clusters) due to changes in the shape of annual hydrograph can be explicitly linked to one another and quantified by changes in IHAs (see **Figure 4** in the revised manuscript). Although approaches such as PCA/EOF are informative, they cannot provide such an opportunity. For example, components in PCA analysis are combinations of several IHAs, whereas we directly link the changes in clusters to individual IHAs and measure the strength of such links using rates of shift and the coefficient of determination. In addition, combining the clustering algorithm with the moving-window technique provides an opportunity to identify the shifts in the regime type in a fully integrated way using a set of specific reference points, i.e., cluster centers in the first time episode. We believe these strengths and added values justify the use of our proposed methodology. We discussed these points in the revised manuscript. Please see **Lines 75-99** in the revised manuscript.

5. *Why use a combination of Kendall's tau and R2 for the attribution work? To me it appears to be a bit convoluted to use two very different metrics (Kendall's tau and R2) for the attribution work. While Kendall's tau operates on ranks and is thus less sensitive to non-linarites or outliers, R2 is in essence a linear metric. As an alternative single metric I could e.g. imagine to rely on Spearmans rank-correlation coefficient together with a simple test of significance thereof.*

**Response:** Many thanks for your comment. We did not combine these two metrics nor the concept behind the Kendall's tau dependence and the coefficient of determination. In fact, we use these two metrics for two different purposes: On the one hand by using Kendall's tau, we identify the sign and significance of dependencies between changes in membership degrees and changes in streamflow characteristics. After the significance of dependence is validated, then on the other hand, we use the coefficient of determination, $R^2$, to quantify how much of the variability in a given set of membership degrees can be described linearly by changes in a specific streamflow characteristic By using these two measures together, we not only provide a formal and robust approach to assess the dependencies between changes in membership degrees and streamflow characteristics (through the use of Kendall's tau with a formal *p*-value), but also we can facilitate quantitative communication of the impact of changes in a specific streamflow characteristic on transition from one regime type to another. We added this clarification in our revised manuscript. Please see **lines 213-218** in the revised manuscript.

6. *Why not just look at trends in time series of monthly means and timing indicators? The analysis revolves around what the authors refer to as Indicators of Hydrologic Alterations (IHA). These are essentially a: The annual mean. b: the mean of each month and c: the timing of low/high flows. While reading I wondered if it would not have been sufficient to simply show maps of the trends of each of these metrics to arrive at the same conclusions?*

**Response:** Many thanks for your comment. There are two separate issues here that we would like to discuss them independently.

First, while we investigated the expected values of annual and monthly mean flows as well as timings of low/high flows, we also looked into the variability of these IHAs during a multi-year timeframe, decadal in our case. While the majority of current literature limit the analysis of change to expected values of IHAs, there are strong evidences, particularly in Canada, revealing that changes in the variability of streamflow characteristics can be important and quite significant– please see Table 4 for some previous findings related to the importance of variability in IHAs and our new findings, showing the variability in IHAs can be as important as the expected values.

Second, while looking at the individual trends in mean and variability of IHAs can be informative, it has certain limitations. Most importantly, looking solely to the trends does not provide any information on how

one regime types evolve from one to another. In addition, looking at individual IHAs or their trends does not necessarily provide any information on how streams group together and are similar/dissimilar to one another. Using our proposed algorithm, we are not only able to group streams into clusters and quantify their belongingness to each group using a unique membership function, but also we know that an increasing trend in membership of one regime type will inevitably translate to decreasing trends in membership values for at least another regime type. Finally, through the use of our proposed approach, it would be possible to formally relate the changes in regime types to changes in the streamflow characteristics. None of these would have been possible by looking only at the simultaneous trends in the individual streamflow characteristics.

7. *Why stratify the analysis along large drainage basins. While I acknowledge the tradition of stratifying the analysis of streamflow data along drainage basins I wonder if this is the ideal choice in this particular instance. The regime classes identified by the authors essentially reflect different climatological regions (e.g. colder, snow dominated vs. warmer, rainfall dominated). Continental-scale drainage basins typically cover large climatic gradients and apart from the case where stations are hydrologically connected (how many of them are?), we would not expect a-priory the drainage basin would have much explanatory power on the climatology. Alternatively, I could imagine an assessment of changes in the underlying climate drivers (e.g. temperature, precipitation) would contribute to a deeper understanding of the associated changes.*

**Response:** Many thanks for your insightful comment. A similar comment was also given by AR1 regarding the suitability of the basin/sub-basin system in discussing/framing our results. Accordingly, we majorly revised our manuscript and considered terrestrial ecozones (Wiken, 1986; Lespinas et al., 2015) as the primary units in which we frame our results. This effort has significantly improved the presentation and interpretation of our results. We have also considered climate regions provided by Environment and Climate Change Canada, but we figured that ecozones provide the most suitable scaling unit for discussing our results. This might be due to the fact that ecozone not only considers climatic factors but also geology, soil characteristics, vegetation, topography, etc. (Wong et al., 2017) that constitute streamflow generation. Just as an example, **Figures 5, 6 and 7** in the revised manuscript show the distributions of the considered RHBN streams as well as the result of our clustering analysis and trend analysis presented across ecozones.

Regarding the hydrological connectivity of the considered RHBN streams, we did a rigorous analysis using HYDAT and National Hydrographic Network data (NHN) provided by Natural Resources Canada to determine the streamflow network and the flow direction at each sub-basin. Accordingly, we realized that there is only one pair of stations (i.e., 01AD002 and 01AD003 located in Saint John- St. Croix sub-basin) that are hydrologically connected. To keep the consistency of our analysis, we exclude 01AD003 from our revised manuscript, so that we have 105 streams that are hydrologically independent from one another. We clarified this in the paper. Please see **line 288 to 290** in the revised manuscript.

Lespinas, F., Fortin, V., Roy, G., Rasmussen, P., & Stadnyk, T.: Performance evaluation of the Canadian precipitation analysis (CaPA). Journal of Hydrometeorology, 16(5), 2045-2064, https://doi.org/10.1175/JHM-D-14-0191.1, 2015.

Wiken, E.B.: Terrestrial Ecozones of Canada. Ecological Land Classification, Series No. 19. Environment Canada. Hull, Quebec. pp. 26, 1986.

Wong, J. S., Razavi, S., Bonsal, B. R., Wheater, H. S., and Asong, Z. E.: Inter-comparison of daily precipitation products for large-scale hydro-climatic applications over Canada, Hydrol. Earth Syst. Sci., 21, 2163–2185, https://doi.org/10.5194/hess-21-2163-2017, 2017.

**Part II. Presentation**

*8. Overall, I found the paper too long, a bit convoluted and therefore cumbersome to read. Some reasons:*

**Response:** Many thanks for your constructive comment. We have rigorously revised our manuscript to address your comment and those raised by AR1. Now that we extended on the methodological aspect of our study, strategically shifted the discussion from basins to ecozones (and accordingly changed our figures) and focused more on key take-home messages, we believe our results are much easier to follow. Please also note that in terms of word count, our manuscript is now reduced by 11%.

*9. The key selling point advertised in the title (i.e. the methodology) is featured in about 10% (4 of 39 pages) of the article and the properties of the methodology (e.g. stability or relation to alternative techniques) are neither assessed and nor discussed.*

**Response:** Many thanks for your comment. While we wanted to be very clear in what we do methodologically and how we do it, we wanted to also be concise and convey the majority of the methodological details through referencing. In the revised version, we indeed extended on explaining our methodology. While our paper has shrinked by 11%, our text in methodology has increased by more than 20%. We extended the explanation related to validation indices and the rationale behind selecting the optimal number of cluster (please see **lines 153-176** in the revised manuscript). We also added a brand new analysis on stability/uncertainty of our approach (**Sect. 5.1**). In addition, we have discussed the pros and cons of the proposed fuzzy-based framework compared to the alternative approaches e.g., EOF/PCA and/or other clustering methodologies in **lines 75-99** in the revised manuscript.

*10. The description of the results is very convoluted and I found it difficult to extract the key message upon first reading. For example, I would value if the results description would focus on overarching patterns/conclusions instead of a diligent, but lengthy description of details.*

**Response:** Many thanks for this very constructive comment. As the unit of our discussion in changed now from drainage basins to ecozones, key messages as well as the general patterns are more obvious in our revised manuscript. Please note that we have changed several figures related to analyses of trends in memberships, shifts in regime types as well as the attributions of regime shifts to the alterations in streamflow characteristics. Accordingly, we have shortened the description of the results and focused only on the key results.

*11. I found figures 8,9,10,11 quite hard to assess on first reading. Would it be possible to summarize these results e.g. in sets of 6 maps (one for each cluster).*

**Response:** Many thanks for your comment. To address you concern, we changed the visualization of the results extensively. In particular, we presented the results of the clustering, trend analysis and regime shifts in sets of 6 maps. Please see the **Figures 6 and 7** in the revised manuscript as well as Figure S6 in the supplement. Regarding Figures 10 and 11, we removed the unnecessary panels, combine the two figures and improve the readability of the figures. Please see **Figure 9** in the revised manuscript. We have also considered a brand-new Sankey diagrams, showing how streamflow regimes in considered RHBN streams transform from one regime to another. Please see **Figure 8** in the revised manuscript. For additional changes made in the presentation of our results, we invite you to also review our responses to the comments #3 and #6 given by AR1.

**Part III. Minor issues**

*12.      There is a significant number of grammatical mistakes (i.e. missing articles) in the paper that need to be resolved by the authors.*

**Response:** Many thanks for your comment. We rigorously edited our paper to avoid grammatical mistakes.

*13.      Text following equation 2c: V (matrix of centroids) is described twice. This indicates that the methods section might have been written in a sloppy manner, raising the question if everything is correct. I did not have the time to check all the indices etc. in detail.*

**Response:** Thank you very much for your thorough review. We cleared this issue in the text. We also double-checked all formulas and their description in the methodology to avoid such mistakes. All other typos and inconsistency issues in the formulas and notations are also taken care of.

*14.      How is the "timing of annual low/high flow" defined? Is it the day of year of the smallest/largest value? If yes: how is the discontinuity between day 365 and day 1 handled?*

**Response:** Many thanks for your comment. The timing is defined on the weekly scale (week 1 to 52). Therefore, for each year a number between 1 and 52 will be chosen, representing the week with highest flow during a given year. We noted that going finer into daily scale can raise issues, particularly in smaller catchments, in which our results become very sensitive to abrupt weather events such as warm spells or rain-over-snow events (Déry et al., 2009).

Déry, S. J., Stahl, K., Moore, R. D., Whitfield, P. H., Menounos, B., and Burford, J. E.: Detection of runoff timing changes in pluvial, nival, and glacial rivers of western Canada. Water Resources Research, 45(4), https://doi.org/10.1029/2008WR006975, 2009.

---

## Author Response (AR2)

**A global algorithm for identifying changing streamflow regimes: Application to Canadian natural streams (1966-2010)**

**Point-to-point Reply to the Editor's and Reviewers' comments**

Masoud Zaerpour, Shadi Hatami, Javad Sadri, and Ali Nazemi

**I.  Reply to the comment made by the Editor**

1. *thanks for being patient. The two reviewers and I find the manuscript much improved and note only minor issues to be resolved now. Please clarify them, in particular the notation issues raised by R1 and please really make an effort to improve the language as noted by R2. Both, clarity and concise writing without unnecessary words will ultimately improve the impact. I am looking forward to the revised version of your manuscript.*

   **Response:** Many thanks for handling our manuscript. We are very pleased that our efforts during the discussion and revision of the paper are recognized by the editor and the two anonymous reviewers. We carefully revised the manuscript to avoid grammatical issues and to have a concise and clear contribution. We are grateful for the comments received from both reviewers and can clearly see the positive impacts of these comments on our final product. We highly appreciate their time, and yours indeed, put selflessly on our paper; and we very much look forward to having this paper out soon.

**II. Reply to comments received from Anonymous Reviewer 1**

1. *The authors are to be congratulated on their revision, which has greatly improved the paper. I was very pleased to see that their analyses now include ecozones, and the paper is much stronger for it. The figures are very attractive.*

   **Response:** We greatly acknowledge the time and effort put by Anonymous Reviewer 1 (AR1) on further evaluation of our manuscript. We are thankful for the constructive comments provided by AR1 and are happy that our revised manuscript was found improved and visually appealing by AR1.

2. *Unfortunately, the results are undermined by the writing. I am providing some suggestions, but the paper would benefit from a thorough editing to tighten the writing, and to remove the many grammatical errors, poor word choices, and excess verbiage.*

   **Response:** Many thanks for your comment. We do acknowledge your concern. We have rigorously edited our paper to have a concise and clear manuscript. We implemented your suggestion throughout the paper. We believe that this comment made us to have a better manuscript now.

   *General comments*
3. *The paper is riddled with cliches:*
   *"to name a few"*
   *"We recognize that"*
   *"Considering the available data"*

*"Looking however at the drainage basins"*
*"First and foremost"*
*"sets the scene"*

*These are unnecessary, and colloquial. Please delete them.*

**Response:** Many thanks for your comment. We edited the paper and revised the word choice as AR1 suggested.

4. *The terms "higher", and "lower", unless you are talking about elevations, should be "greater" or "larger" and "smaller"*
**Response:** Many thanks for your comment. This is taken care of now throughout the manuscript.
*.*

**Detailed comments**
5. *Line 27*
*"early settlements"*
*These are things, not points in time. I think you mean "the times of the first human settlements"*
**Response:** Many thanks for your comment. We revised the sentence.

6. *L 29*
*"reveal"*
*This word is used far too often, and incorrectly.*
*I think that "indicate" is what you mean.*
**Response:** Many thanks for your comment. This is taken care of throughout the manuscript.

7. *L 30*
*"some"*
*I think that "others" is more useful*
**Response:** Done.

8. *L 50*
*"it is"*
*Disagreement in number. Should be "are"*
**Response:** Done.

9. *L 60*
*"relativity"*

   *This is rather confusing in a scientific paper, particularly when you refer to "time and space"!*
*It's also a bit vague.*
*What is relative to what?*
**Response:** Many thanks for your comment. It is substituted with "subjectivity".

10. *L 77*
*"some of which with"*

*Replace "with" by "have"*
**Response:** Done.

11. *"dynamics"*
   *I'm not sure what this means*
   **Response:** the dynamic is now replaced by "trends"

12. *L 78*
   *"with characteristics"*
   *Should be*
   *"of characteristics"*
   **Response:** Done.

13. *L 81*
   *"makes"*
   *I think that "creates" would be better.*
   **Response:** Agreed**.** Done.

14. *L 126*
   *"in a way"*
   *Would advise inserting "such" after "in"*
   **Response:** Done.

15. *L 142, eq. 2A*
   *It would be nice to have a verbal description of the objective function. Does it have a reference?*
   **Response:** Many thanks for your comment. The description and the reference are added. Please see **line 132.**

16. *L 177*
   *I was confused in this section.*
   *The meaning of the baseline isn't really explained very well.*
   **Response:** Many thanks for your comment. The explanation is added to the **line 166** in the revised manuscript.

17. *L 190*
   *"sum of memberships in each timeframe is one"*
   *I suggest that you use "1" as "one" could be part of a phrase, like*
   *"one of the ...."*
   **Response:** Done.

18. *L 191*
   *"one or more clusters".*
   *This is confusing. Shouldn't it be "other clusters"?*

**Response:** You are right**.** Done.

*19. L 218*
   *"transparency"*
   *How it this method more transparent than others?*
   Response: We revised the sentence to clarify better what we mean. Please see **line 212** in the revised manuscript.

*20. L 219*
   *"is tied up with attribution"*
   *Could be shortened to "attributed"*
   **Response:** Done.

*21. L 221*
   *"Black and red lines"*
   *Please add a leading "The"*
   **Response:** Done.

*22. "expected annual hydrographs"*
   *What are these – how are the values expected? Please explain.*
   Response: Clarified in the text. Please see revised version **line 215**.

*23. L 236, 3 Case study and data*
   *A lot of the following information could be summarised in a table.*
   Response: This can be true and we used tables and other elements in the Supplement. However, we believe our paper should have a concise descriptive overview on Canadian drainage basins and ecozones particularly for non-Canadian readers. We keep the description but made an effort to shorten and restructure to improve the flow.

*24. L 237*
   *"With the total catchment area equivalent to"*
   *Replace with "With a total catchment area "*
   **Response:** Done.

   *"roll coast to coast to coast and during their journey"*
   *Unnecessary, please delete*
   Response: Done.

*25. L 245*
   *"largest water bodies"*
   *Not a very good description. Perhaps you could say some of the largest lakes other than the Great Lakes.*
   **Response:** Agreed. It is now revised.

*26. L 246*
   *"Mackenzie"*
   *Should be preceded by "The"*
   **Response:** Done.

*27. L 250*
  *"While drainage basins"*
  *"While" is usually used to mean "at the same time as".*
  *I would suggest using "Although"*
  **Response:** Done.

  *"in which the streamflow is traced from headwaters to oceans"*
  *Unnecessary, please delete*
  **Response:** Done.

*28. L 254*
  *"patch of terrestrial land"*
  *Unless we are talking about other planets, "land" implies "terrestrial"*
  *Suggest replacement with*
  *"terrestrial region"*
  **Response:** Done.

  *"distinc"*
  *Misspelled – should be "distinct"*
  **Response:** Done.

*29. L 256*
  *"only major Canadian region outside the Rockies"*
  *Really? Isn't this contradicted by the existence of every other ecozone?*
  **Response:** Many thanks for your comment. This was a typo. We meant the only major Canadian mountainous region other than the Rockies. It is now corrected.

*30. L 340*
  *"glacial-fed or lake-dominated streams"*
  *This would be a good place to add a statement on the effect of storage to the slow responses of these streams.*
  **Response:** Many thanks for your comment. A description is added to explain how storage of lakes lead to slow response of the streams. Please see **line 329** in the revised manuscript.

*31. L 345*
  *"other processes such as fill and spill"*
  *Agreed. However the process should probably be explained for readers not familiar with it.*
  **Response:** A short description of fill and spill is added. Please see **line 335** in the revised manuscript.

*32. L 381*
  *"belongingness to"*
  *This isn't the right word in this context, as it refers to an emotional state. I would use "member of"*
  **Response:** Done.

*33. L 386*
  *"less"*
  *Should be "fewer"*
  **Response:** Done.

*34. L 413*
   *"vibrant"*
   *Not a good word to use as its meaning is unclear in this context.*
   **Response:** Many thanks for your comment. It is now substituted by "varied"

*35. L 434*
   *"percentage"*
   *Looks like the fraction to me*
   **Response:** Many thanks for your comment. It is now substituted by the fraction.

*36. L 436*
   *"second cell from right" – I would say "left hand cell"*
   *"first cell from the right" – I would say "right hand cell"*
   **Response:** Done.

*37. L 450 Figure 9*
   *I couldn't tell what the sizes of the squares mean – are they a function of some value?*
   *If not, it would make the colour differences more visible if the squares were the same size.*
   **Response:** Many thanks for your comment. Yes, it is a function of values of $R^2$. The smaller $R^2$ corresponds to the smaller the size of the square. This is for better visibility of the larger values and avoiding a busy figure.

*38. L 479*
   *"choosing"*
   *I would say "selecting"*
   **Response:** Done.

*39. L 485*
   *"due to the insufficiency"*
   *Of what?*
   **Response:** due to the insufficiency in number of data points. It is now corrected in the text.

*40. L 494*
   *"not significant changes"*
   *I may have missed this – where was the significance test?*
   **Response:** The word significant was misleading. It is now revised.

*41. L 496*
   *Figure 10*
   *What are Dimension 1 and 2? I know that they are referred to in line 489, but it would be nice to see an explanation.*
   **Response:** A short description added. Please see **line 483** in the revised manuscript.

*42. L 519 - 5.2 Validation in unseen streams*

*I think that "unseen" is a bit confusing. Perhaps you could refer to the out of sample streams.*
**Response:** Done. It is now substituted by the out of sample streams.

43. *L 523*
*"because the majority of annual streamflow volume is contributed from mountainous headwaters outside of Prairies"*
*This is true of the large rivers, as well the smaller streams you examine in the extreme western portion of the ecozone (such as the Waterton River near Waterton Park and Belly River near Mountain View), however it is not true of the other prairie streams that you feature.*
**Response:** Many thanks for your comment. You are totally right. Other factors are also added to include other prairie streams. Please see the **lines 521-525** in the revised manuscript.

44. *"fact that large proportion of the land does not contribute into the streamflow"*
*This needs a citation. Also, it is not true for some basins in the region, and is not always true for the remainder, as the contributing fractions of these basins vary over time.*
**Response:** Many thanks for your comment. The references are added. As you said, it may not be true for all basins in the Prairies, so the sentence is revised. Please see the **line 524** in the revised manuscript.

45. *Another issue is that many Prairie streams only have seasonal records, which can mess up analyses requiring entire years of records.*
**Response:** Many thanks for your comment. We only considered streams with continuous data records for the analyses.

46. *L 529*
*"Waterton River near Waterton Park (S69) and Belly River near Mountain View (S70)"*
*Since you are referencing the streams in Figure 12 by their WSC gauge IDs, you should also list the IDs here.*
**Response:** Many thanks for your comment. We also added the gauge IDs to the text.

47. *L 532*
*"are sorted from the east to west"*
*I would use "ordered" rather than "sorted".*
**Response:** Done.

48. *L 545*
*"a fully algorithmic framework"*
*I don't understand what this term means*
**Response:** It is now revised.

49. *Finally, I was disappointed to see that all of the analyses were done using a closed-source language. This is particularly troubling because the analyses include significance tests, where the value of a calculated statistic is compared to a fixed threshold (i.e. $p = 0.05$), resulting in acceptance or rejection of a null hypothesis. A small error in the statistic can therefore result in a very large effect, and we can have no understanding of how the statistic was calculated. I would encourage the authors to use open-source languages in the future for their analyses.*

**Response:** Many thanks for your comment. Totally agreed. We surely consider doing the analyses in an open-source language in the future.

**III. Reply to comments received from Anonymous Reviewer 2**

1. *The paper presented by Zaerpour and colleagues has much improved and I second the author's effort to increase the clarity of their Work while making the text more compact at the same time. Overall, I very much appreciate the effort of the authors to put their methodological development into context. I also welcome that the case study now revolves around the analysis of climatologically distinct ecozones in Canada. With these significant additions, I find that the paper is almost ready for publication in HESS.*

   **Response:** We greatly acknowledge the time and effort Anonymous Reviewer 2 (AR2) put on evaluating our revised manuscript. We are very happy that AR2 found our revisions effective!

   *Nonetheless, a few comments remain:*

2. *P5, L 523: It is not clear what X and Y are. I suspect X = mean and Y = sd but making this explicit would help.*
   **Response:** Many thanks for your comments. You are totally right. X and Y are the expected mean and variance of 15 indicators. We added an explanation to **line 421**.

3. *Notation of Equation 2a and related text: I suggest to use a capital C for denoting the cluster number to indicate that this is not an index. Similarly: consider to replace the index k with small c, which might increase the readability.*
   **Response:** Many thanks for your comment. Totally agreed. We revised notations in Eq. 2a and elsewhere, accordingly. It reads better now.

4. *Mix of Kendall-statistics and R2 for the attribution work: I still find it unconventional to first rely on a rank-statistic for identifying significance and then use a conventional R2 for quantifying the strength of the association. To me it would be more natural to either (a) pair the Kendall statistics with the (squared) rank correlation coefficient or (b) rely on the (squared) Pearson correlation coefficient alongside a standard test for significance for this metric. Note that rank-correlations (e.g. Spearman's coefficient) should indicate even higher degrees of association in case the data have a non-linear relation. Note, however, that I am confident that the choices made here will not have a significant impact on the conclusions.*

   **Response:** Many thanks for your comment. We used $R^2$ to determine how much variability in the degree of membership can be described by the variability of a given streamflow characteristic. However, to address your concern we repeated the calculation using squared Kendall tau measure which is rank-based measure of dependence. As you can see in the following Figure the result does not change significantly. In some cases, however, the degree of association using Kendall tau measure increase compared to the $R^2$ as you mentioned, which means we see a stronger link. We added the Fig. 1 shown below (Fig. S9 in the supplement) and the discussion related to this analysis on the sensitivity of the attribution results to the choice of association measure in **lines 509-514** to the discussion Section.

[Figure]

**Figure 1.** Dominant regime shifts across 105 RHBN streams in Canada attributed to the first and second moments of the 15 IHAs considered. Shades of red and blue show the values of squared Kendall's tau between changes in streamflow features and degrees of membership. The dominant regime shift at each stream is identified by the color scheme described in the legend. Streams are grouped in ecozones and ordered from low (top) to the high (bottom) elevations.

5. *Fig. 1: indicating what NSF_1 or NSF_2 are in the caption would make it easier to understand the concepts.*
   **Response:** Many thanks for your comments. We think you meant Fig.3. If so and for the sake of clarification, an explanation is added to the caption. Please see **line 200**.